# D-CPT Law: Domain-specific Continual Pre-Training Scaling Law for Large Language Models

**Haoran Que*[1], Jiaheng Liu*[†,1], Ge Zhang*[3,7], Chenchen Zhang[2], Xingwei Qu[4,7], Yinghao Ma[5,7], Feiyu Duan[1], Zhiqi Bai[1], Jiakai Wang[2], Yuanxing Zhang[2], Xu Tan[7], Jie Fu[6], Jiamang Wang[2], Lin Qu[2], Wenbo Su[1], Bo Zheng[1]**

[1]Taobao & Tmall Group of Alibaba, [2]Alibaba Group, [3]University of Waterloo
[4]University of Manchester, [5]QMUL, [6]HKUST, [7]M-A-P
{quehaoran.qhr, ljh411989}@taobao.com, gezhang@umich.edu

## Abstract

Continual Pre-Training (CPT) on Large Language Models (LLMs) has been widely used to expand the model's fundamental understanding of specific downstream domains (e.g., math and code). For the CPT on domain-specific LLMs, one important question is how to choose the optimal **mixture ratio** between the general-corpus (e.g., Dolma, Slim-pajama) and the downstream domain-corpus. Existing methods usually adopt laborious human efforts by grid-searching on a set of mixture ratios, which require high GPU training consumption costs. Besides, we cannot guarantee the selected ratio is optimal for the specific domain. To address the limitations of existing methods, inspired by the Scaling Law for performance prediction, we propose to investigate the Scaling Law of the Domain-specific Continual Pre-Training (**D-CPT Law**) to decide the optimal mixture ratio with acceptable training costs for LLMs of different sizes. Specifically, by fitting the D-CPT Law, we can easily predict the general and downstream performance of arbitrary mixture ratios, model sizes, and dataset sizes using small-scale training costs on limited experiments. Moreover, we also extend our standard D-CPT Law on cross-domain settings and propose the **Cross-Domain D-CPT Law** to predict the D-CPT law of target domains, where very small training costs (about 1% of the normal training costs) are needed for the target domains. Comprehensive experimental results on six downstream domains demonstrate the effectiveness and generalizability of our proposed D-CPT Law and Cross-Domain D-CPT Law.

## 1 Introduction

**Continual Pre-Training (CPT)** is an essential part of training better Large Language models (LLMs). In this work, we mainly focus on Domain-specific CPT (**D-CPT**), which aims to enhance the fundamental understanding abilities of the specific downstream domains and has been widely used in existing works [49, 41, 30]. In practice, for D-CPT, we usually need to collect high-quality domain-corpus to enhance the downstream performance and general-corpus to mitigate catastrophic forgetting on the general abilities [13, 40, 54, 47, 32, 20, 53]. Therefore, how to determine the data composition or mixture ratio of the domain-corpus and general-corpus plays an important role in producing well-performed domain-specific LLMs. Besides, grid-searching on the mixture ratios requires heavy GPU consumption costs, and we cannot always obtain the optimal ratio under limited GPU usage. Recently, Scaling Law has been widely used for performance prediction [31, 27, 43, 26],

---

* First three authors contributed equally.

† Corresponding Author: Jiaheng Liu.

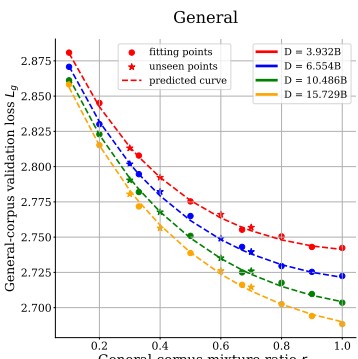
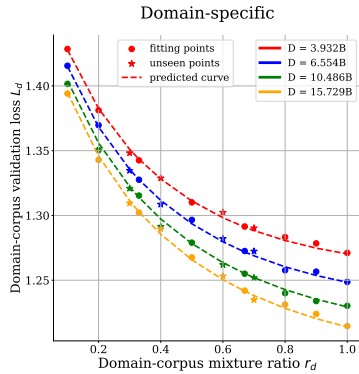

Figure 1: Illustration of the performance of **D-CPT Law**. (*Left*): The curves show the relationship between $L_g$ and $r_g$ under different dataset sizes $D$ for Qwen1.5-1.8B model. CPT data are a mixture of **code-corpus** and general-corpus. Here, $L_g$ represents the loss on the general-corpus validation set, while $r_g$ indicates the percentage of the general corpus in the training data. The dashed curves denote the curves predicted by D-CPT Law, circular markers and star markers are fitting data points and unseen validation points, respectively. (*Right*): These curves are the corresponding results between the code-corpus validation loss $L_d$ and the percentage of the code-corpus data $r_d$.

which can be used to find the optimal dataset size and model size under the given GPU consumption costs. Therefore, *for D-CPT, can we find the optimal mixture ratio in the training corpus using the Scaling Law to enhance the performance of domain-specific tasks?*

To address the above question, in this work, we investigate the Scaling Law of D-CPT and propose the **D-CPT Law** to find the optimal mixture ratio with limited training costs for LLMs with different sizes. Specifically, inspired by the robust predictive ability of Scaling Law across various scales, we first perform experiments under diverse mixture ratios and several relatively small model and data scales. Following the Chinchilla Scaling Law, we then introduce the mixture ratio $r$ into the D-CPT Law, where the parameterization is defined as follows:

$$L(N, D, r) = E + \frac{A}{N^\alpha} + \frac{B \cdot r^\eta}{D^\beta} + \frac{C}{r'^\gamma}, \text{ where } r' = r + \epsilon, \tag{1}$$

where $\epsilon$ is used to guarantee the stability of $L$ when $r$ near zero. Based on Equation 1, for a model with model size $N$, dataset volume $D$ and mixture ratio $r$, we can accurately predict the validation loss $L$. Note that when $r$ denotes the domain-corpus mixture ratio $r_d$, $L$ means domain-corpus validation loss $L_d$. Similarly, general-corpus validation loss $L_g$ also follows the law relationship with the general-corpus mixture ratio $r_g$. To illustrate our D-CPT Law clearly, as shown in Figure 1, we take the code domain as an example and provide the fitting results on the general and domain-specific settings, where we validate the fitting accuracy on different mixture ratios under a model with different dataset sizes $D$. Our main contributions are summarized as follows:

(1). To show the effectiveness and generalizability of D-CPT Law, we perform extensive experiments using model sizes from 0.5B to 4B parameters, dataset sizes from 0.1B to 26B tokens, and mixture ratios from 0 to 1. The experiments show that the D-CPT law exhibits a high fitting accuracy with Huber loss [28] lower than 0.02 and $R^2$ [9] greater than 0.97. Besides, experiments on generalizability show that D-CPT Law not only inherits model size and dataset size generalizability following previous Scaling Law, but also precisely predicts performance for different mixture ratios.

(2). Despite the effectiveness in an in-domain setting, where we fit the D-CPT Law based on data points from one downstream domain, we also apply our D-CPT Law in the cross-domain setting, which denotes that we use the data points from multiple domains to predict the performance of unseen domains. Specifically, we first introduce the **Domain-specific Learnable Coefficient (DLC)** to denote the domain-specific parameter of each domain and integrate the DLC into the D-CPT Law. We name this new law as **Cross-Domain D-CPT Law**. In this way, if we can obtain the DLC of a new domain, we can easily derive the D-CPT Law for this new domain. In our experiments, we fit the Cross-Domain D-CPT Law using data points from 4 domains and apply the Cross-Domain

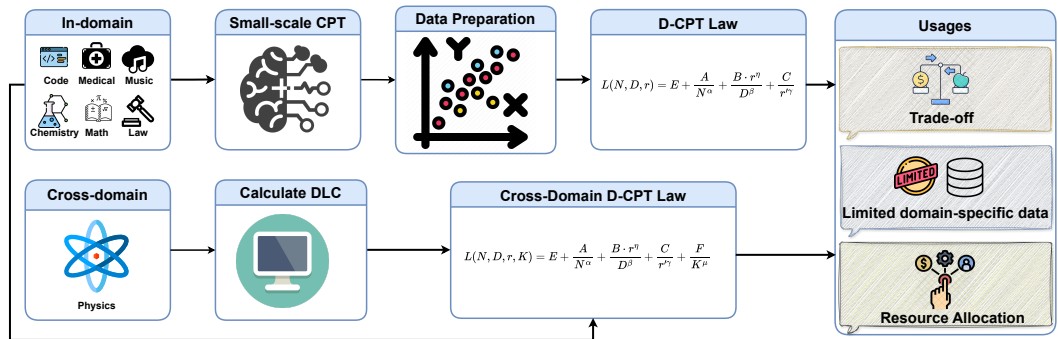

Figure 2: Illustration of **D-CPT Law** and **Cross-Domain CPT-Law** pipeline. (*Upper*): In D-CPT Law, we first collect domain-corpus and general-corpus, and conduct experiments under a small-scale experimental setup to gather empirical data points to fit the D-CPT Law. After that, we can predict the model's performance in large-scale experimental settings. (*Lower*): In Cross-Domain CPT-Law, for an unseen downstream domain, like Physics, we can calculate its Domain-specific Learnable Coefficient value and incorporate it into the fitted Cross-Domain D-CPT Law to derive the D-CPT Law for this new domain. Based on the D-CPT Law, we introduce three application scenarios: optimal mixture on the trade-off between general and domain-specific abilities, optimal mixture for limited domain-specific data, and resource allocation in Section 4.3.

D-CPT Law to predict the remaining 2 domains. The results show that DLC can represent the specific information for each downstream domain well, enabling efficient and effective fitting performance for the cross-domain setting and significantly reducing training costs for new domains.

(3). To show the real-world usages of the D-CPT Law, we apply our D-CPT Law on three important scenarios: optimal mixture on the trade-off between general and domain-specific abilities, optimal mixture for limited domain-specific data, and resource allocation setting in Figure 2 (Details are provided in Section 4.3).

## 2 Background

Following previous work [43], we categorize the objectives of Scaling Law as *Allocation* and *Return*. Specifically, (1). *Allocation*: What is the optimal allocation of model size $N$ and dataset size $D$ given a fixed compute budget? (2). *Return*: What is the expected return on incremental resources?

The first objective on *Allocation* is as follows:

$$\underset{N,D}{\operatorname{argmin}} L(N, D) \quad \text{s.t.} \quad \text{FLOPs}(N, D) = C. \tag{2}$$

In Equation 2, given a fixed compute budget $C$, the objective is to find the optimal model size $N$ and dataset size $D$ that minimize the loss. The second objective on *Return* fundamentally depends on the generalizability of Scaling Law to accurately predict beyond the fitting data points.

**Chinchilla Scaling Law** Hoffmann et al.[27] propose a parameterization as follows:

$$L = E + \frac{A}{N^\alpha} + \frac{B}{D^\beta}, \tag{3}$$

where $\{E, A, B, \alpha, \beta\}$ are fitting parameters (See Appendix D for more details).

## 3 Methods

In Figure 2, D-CPT Law aims to investigate the behaviors of the Domain-specific Continual Pre-Training scenario with respect to different mixture ratios, and the objective of the D-CPT Law is to analyze an appropriate parameterization of law that represents the relationship of validation loss $L$

with respect to model size $N$, dataset size $D$, and mixture ratio $r$. In this section, we first discuss the D-CPT Law in the in-domain setting (Section 3.1), where the fitting and testing data points are from the same domains. Then, we propose to adapt D-CPT Law to the cross-domain setting (Section 3.2), where the fitting and testing data points are from multiple domains, and introduce the Cross-Domain D-CPT Law, where a new term Domain-specific Learnable Coefficient (DLC) is used.

## 3.1 D-CPT Law

As the training data includes a mixture of general-corpus and domain-corpus, we introduce two mixture ratios (i.e., general-corpus mixture ratio $r_g$ and domain-corpus mixture ratio $r_d$). Correspondingly, we define two validation losses (i.e., general-corpus validation loss $L_g$ and domain-corpus validation loss $L_d$). Therefore, we can derive two D-CPT Laws (i.e., $L_g(N, D, r_g)$ and $L_d(N, D, r_d)$). For convenience, we directly use $r$ and $L(N, D, r)$ as default notations for D-CPT Law. Besides, in Appendix D, Chinchilla Scaling Law provides greater interpretability and clarity when compared to OpenAI Scaling Law. Thus, we choose Chinchilla Scaling Law as the foundational parameterization for D-CPT Law. In addition, since Scaling Law aims to fit data points, their parametric forms should be intrinsically related to the observed trends in the data points. Based on previous works and data trends with varying $N$, $D$ and $r$, we have summarized 4 essential requirements for D-CPT Law:

- **Adaptability**: D-CPT Law is valid for values of $r$ between 0 and 1.
- **Explicit trends**: Based on the results across varying values of $N$, $D$, and $r$, we observed the following explicit trends of data points:

$$\frac{\partial L}{\partial N} < 0, \quad \frac{\partial L}{\partial D} < 0, \quad \frac{\partial L}{\partial r} < 0. \tag{4}$$

  The first two trends are consistent with the previous Chinchilla Scaling Law, and the third trend also has an intuitive explanation. A larger $r$ indicates a higher proportion of valid corpus in the training corpus, leading to a lower $L$. Details are provided in Appendix E.1.

- **Implicit trends**: We further discover inherent connections between $r$, $D$ and $L$ as follows:

$$\frac{\partial^2 L}{\partial D \partial r} < 0. \tag{5}$$

  For detailed explanations, please refer to the Appendix E.2.

- **Consistency**: When $r$ is fixed, the D-CPT Law is supposed to transform into the Chinchilla Scaling Law. In this way, D-CPT Law can inherit the excellent features of Chinchilla Scaling Law and address the issues on resource allocation discussed in Section 2.

To satisfy these requirements, we have compared multiple parameterizations in Section 4.2 and eventually, we propose the parameterization as follows:

$$L(N, D, r) = E + \frac{A}{N^\alpha} + \frac{B \cdot r^\eta}{D^\beta} + \frac{C}{r'^\gamma}, \quad \text{where} \quad r' = r + \epsilon. \tag{6}$$

In Equation 6, $\{E, A, B, C, \alpha, \beta, \gamma, \eta, \epsilon\}$ are the fitting parameters and we use L-BFGS[34] to fit the D-CPT Law due to its suitability for large-scale optimizations. As shown in Section 4.2, the Equation 6 can accurately fit the trends of data points at any scale and demonstrates strong performance in both effectiveness and generalizability. Besides, it effectively meets the aforementioned 4 requirements. (Please see Appendix E.3 for more details on the mathematical derivation.)

## 3.2 Cross-Domain D-CPT Law

Apart from the in-domain setting for D-CPT Law, we also investigate the cross-domain setting and extend the D-CPT Law to the Cross-Domain D-CPT Law, which aims to reduce the training costs of the D-CPT Law significantly. Specifically, although the D-CPT Law collects data points using small LLMs, the GPU resource and time costs are still relatively substantial, which limits the applications of the Scaling Law. Therefore, in our Cross-Domain D-CPT Law, we first define the Domain-specific Learnable Coefficient (DLC) $K$ for each domain, which measures the learnability

for a specific domain[1] (See Section 4.4 for more details.). Then, we incorporate the $K$ into the D-CPT Law and obtain the Cross-Domain D-CPT Law, which is defined as follows:

$$L(N, D, r, K) = E + \frac{A}{N^\alpha} + \frac{B \cdot r^\eta}{D^\beta} + \frac{C}{r'^\gamma} + \frac{F}{K^\mu}, \quad \text{where} \quad r' = r + \epsilon. \quad (7)$$

In Equation 7, $\{E, A, B, C, F, \alpha, \beta, \eta, \gamma, \epsilon, \mu\}$ are fitting parameters. Thus, for an unseen domain, we only need to calculate the DLC with modest costs, which substantially increases the domain generalizability of D-CPT Law. Besides, Cross-Domain D-CPT Law has the following features:

- **Uniformity**: Once we calculate the $K$ value of an unseen domain, we can convert Cross-domain D-CPT Law into normal D-CPT Law as follows:

$$L(K = K_0) = E_0 + \frac{A}{N^\alpha} + \frac{B \cdot r^\eta}{D^\beta} + \frac{C}{r'^\gamma}, \quad \text{where} \quad E_0 = E + \frac{F}{K_0^\mu}, \quad r' = r + \epsilon.$$

  Therefore, the Cross-Domain D-CPT Law inherits all features of the D-CPT Law,

- **Monotonicity**: $K$ denotes the learnability of a specific domain, which aligns with the intuition that a more learnable domain yields lower validation loss. Meanwhile, the Cross-domain D-CPT Law confirms a monotonic decrease with respect to $K$, i.e.,

$$\frac{\partial L}{\partial K} = -\frac{\mu F}{K^{\mu+1}} < 0. \quad (8)$$

After confirming the parameterization of Cross-domain D-CPT Law, it is essential to identify a representation for $K$ that accurately quantifies a domain's learnability. The representation of $K$ is supposed to be **accessible**, **distinct** and **robust**. Specifically, first, "Accessible" denotes that it is easy to obtain for an unseen domain with low costs. Second, "Distinct" indicates that $K$ values must exhibit significant variance across domains to ensure fitting accuracy and maintain clear distinctions between domains. Third, "Robust" means that the representation of $K$ enhances the effectiveness and generalization ability of Cross-domain D-CPT Law. In Section 4.4, we compare several variants of the representations of $K$, and the final representation is determined as follows:

$$K = \frac{w_1}{k_1} + w_2 \times k_2, \quad (9)$$

where $w_1$ and $w_2$ are fitting parameters, $k_1$ represents the initial validation loss for an unseen domain, and $k_2$ denotes the rate of decline in validation loss.

## 4 Experiments

### 4.1 Experimental Setup

**Data Setup** To verify the effectiveness and generalizability of D-CPT Law and Cross-Domain D-CPT Law, we have prepared the 6 different downstream domains, which include Code [39], Math [5], Law [24], Chemistry [10], Music [44] and Medical [33]. For general corpus, we use Dolma [48]. All the tokens of these training datasets are sufficient, so the experiments are not performed under a data-constrained setting. Besides, we build a high-quality and held-out validation set for each domain. (See Appendix F.1 for more details.)

**Model Setup** We use the Qwen-1.5 series due to its robust performance in both English and Chinese [6]. Furthermore, Qwen-1.5 has multiple open-sourced and well-performed pre-training base models. Specifically, we select Qwen-1.5-0.5B, Qwen-1.5-1.8B, and Qwen-1.5-4B as our base models to perform the continual pre-training for multiple downstream domains.

---

[1]Note that we assume lower DLC means easier to learn for the corresponding domain.

Table 1: Mean performance across five parameterizations over six domains. "G" and "D" denote general and downstream domains. Detailed results on all domains are shown in Appendix J.

| Parameterization | Huber loss ↓ | | $R^2$ ↑ | | # fitting parameters |
|---|---|---|---|---|---|
| | G | D | G | D | G/D |
| $L_1$ | 0.0064 | 0.0169 | 0.994467 | 0.976700 | 8 |
| $L_2$ | 0.0050 | 0.0166 | 0.996483 | 0.978283 | 9 |
| $L_3$ | **0.0048** | **0.0157** | **0.996750** | **0.979633** | 9 |
| $L_4$ | 0.0066 | 0.0160 | 0.993567 | 0.978367 | 8 |
| $L_5$ | 0.0328 | 0.0438 | 0.9496 | 0.9512 | **6** |

**Training Setup**   We follow Chinchilla [27] to fix model sizes and vary the number of training tokens for data point collection. Specifically, we test the validation loss every 1,000 steps [2] and the total training steps are 200k. Then, we establish 9 mixture ratios between general-corpus and domain-corpus as follows: {0:10, 1:9, 2:8, 3.3:6.7, 5:5, 6.7:3.3, 8:2, 9:1, 10:0}. Note that all experiments are conducted with the same learning rate schedule (Hyperparameters can be found in Appendix F.2).

**Metrics**   Following [44, 43, 27], we use validation loss as the performance indicator. To compare various parameterizations, we follow [28, 44] to utilize the $R^2$ and Huber loss as evaluation metrics. Specifically, first, the coefficient of determination (i.e., $R^2$) indicates the fitting quality and typically ranges from 0 to 1, where a higher value means better explanatory power of the regression model. Second, Huber loss combines the properties of mean squared error and mean absolute error, which is particularly useful for regression with outliers. Similarly, Huber loss also assesses the fit qualities of different parameterizations, where lower Huber loss shows better fitting performances.

## 4.2   D-CPT Law

In Section 3.1, an ideal parameterization should meet four requirements (i.e., adaptability, explicit trends, implicit trends, and consistency), and we define the following five parameterizations:

$$L_1 = E + \frac{A}{N^\alpha} + \frac{B}{D^\beta} + \frac{C}{r'^\gamma}, L_2 = E + \frac{A}{N^\alpha} + \left(\frac{B}{D^\beta} + \frac{C}{r'^\gamma}\right)^\eta, L_3 = E + \frac{A}{N^\alpha} + \frac{Br^\eta}{D^\beta} + \frac{C}{r'^\gamma},$$

$$L_4 = E + \frac{A}{N^\alpha} + \frac{B \cdot b^r}{D^\beta} + \frac{C}{c^r}, \quad L_5 = E + \frac{A}{N^\alpha} + \frac{B}{(rD + (1-r)\sigma)^\beta},$$

where $\{N, D, r\}$ are variables and others are learned parameters fitted by L-BFGS algorithm[34] which is the same as Chinchilla Scaling Law.

**Effectiveness**   As shown in Table 1, we present the performance of five different parameterizations. In effectiveness settings, we use entire data points for fitting with the aim of validating the effectiveness of various parameterizations. In Table 1, we observe that although $L_5$ has the fewest fitting parameters, its performance is significantly less impressive compared to the others. $L_1$ and $L_4$, having relatively fewer fitting parameters, still fall short in performance compared to $L_2$ and $L_3$. Moreover, $L_1$ fails to meet the requirements for implicit trends, while $L_4$ does not satisfy the explicit trends requirement. Finally, the results of $L_2$ and $L_3$ are comparable, but $L_2$ does not meet the requirements for consistency. Therefore, we choose $L_3$ for D-CPT Law. Moreover, Figure 3 shows the robust effectiveness of $L_3$ across varying dataset sizes, mixture ratios, model sizes, and domains.

**Model Size Generalizability**: Our main experiments focus on 3 model sizes: 0.5B, 1.8B, and 4B. We use 3-fold cross-validation to evaluate the model size generalizability of D-CPT Law, and the average results across domains are shown in Table 2. For example, we fit D-CPT Law with data points from 0.5B, and 1.8B and evaluate the Huber loss and $R^2$ for 4B. In Table 2, we observe that D-CPT Law can generalize well across model sizes and $L_3$ shows the best performance. Besides, we conduct experiments on the unseen 7B size (i.e., Qwen-1.5 7B), and observe that D-CPT Law can accurately predict the general-corpus validation loss with a general-corpus mixture ratio of 0.2 in Figure 4.

---

[2]Training steps to number of training tokens is as follows: $t$ steps $= t \cdot 64 \cdot 2,048 \cdot 10^{-9}$.

**Dataset Size Generalizability**: Our main experiments cover dataset sizes from 0.1B to 26B tokens, and we also utilize a 3-fold cross-validation approach. The data points are uniformly divided into three segments, with 2/3 used for fitting the model and the remaining 1/3 for testing. In Table 3, we report the average results across domains, and observe that $L_3$ shows notably enhanced dataset size generalizability.

Figure 4: $L_g$ with respect to $D$, domain-corpus is code, $r_g = 0.2$, $N = 7B$.

Table 2: Model Size Generalizability.

| Parameterization | Huber loss↓ | | $R^2$ ↑ | |
| --- | --- | --- | --- | --- |
| | G | D | G | D |
| $L_1$ | 0.0055 | 0.0172 | 0.9521 | 0.9366 |
| $L_2$ | **0.0047** | 0.0171 | 0.9663 | 0.9420 |
| $L_3$ | 0.0049 | **0.0166** | **0.9711** | **0.9516** |
| $L_4$ | 0.0054 | 0.0168 | 0.9680 | 0.9453 |
| $L_5$ | 0.0105 | 0.0578 | 0.6835 | 0.8257 |

Table 3: Dataset Size Generalizability.

| Parameterization | Huber loss↓ | | $R^2$ ↑ | |
| --- | --- | --- | --- | --- |
| | G | D | G | D |
| $L_1$ | 0.0065 | 0.0098 | 0.9450 | 0.9069 |
| $L_2$ | 0.0054 | 0.0123 | 0.9352 | 0.8909 |
| $L_3$ | **0.0038** | 0.0096 | **0.9865** | **0.9126** |
| $L_4$ | 0.0084 | **0.0093** | 0.9126 | 0.9037 |
| $L_5$ | 0.1212 | 0.0167 | 0.8686 | 0.8783 |

Table 4: Mixture ratio Generalizability.

| Parameterization | Huber loss↓ | | $R^2$ ↑ | |
| --- | --- | --- | --- | --- |
| | G | D | G | D |
| $L_1$ | 0.0022 | 0.00679 | 0.9950 | 0.9673 |
| $L_2$ | 0.0021 | 0.00695 | 0.9957 | 0.9672 |
| $L_3$ | **0.0019** | 0.00673 | **0.9964** | **0.9717** |
| $L_4$ | 0.0049 | **0.00670** | 0.9797 | 0.9579 |
| $L_5$ | 0.0094 | 0.0256 | 0.9570 | 0.8434 |

**Mixture ratio Generalizability**: We apply the k-fold cross-validation method across various parameterizations. Specifically, we select 7 out of 9 mixture ratios for fitting and the remaining for testing, resulting in 36 experiments per domain. For simplicity, we show average results across domains in Table 4, and observe that that $L_3$ still shows significantly better performance on mixture ratio generalizability. Besides, in Figure 1, we observe that our D-CPT Law has well-performed generalizability on unseen mixture ratios.

### 4.3 Usages of D-CPT Law

**Usage 1: Trade-off between general and domain-specific abilities** For D-CPT, training data is a mixture of general and domain-specific data, where $r_g$ and $r_d$ denote the corresponding proportions, respectively. In D-CPT Law, when $r_g$ increases, the $L_g$ will decrease and $L_d$ will increase, indicating a trade-off between the general and domain-specific abilities of LLM. Fortunately, D-CPT Law can identify the optimal mixture ratio under any trade-off scenario. Specifically, we assume that an LLM with parameter size $N_0$, it exhibits general-corpus validation loss of $L_g^0$ and domain-corpus validation loss of $L_d^0$ before continual pretraining. After mixing training data size of $D_0$ with a ratio $r_d$ of domain-specific data and $1 - r_d$ of general data, we obtain general-corpus validation loss $L_g$ and domain-corpus validation loss $L_d$ after D-CPT. Then, we can identify the optimal mixture ratio while limiting the decline in the model's general abilities within a threshold $T$ as follows:

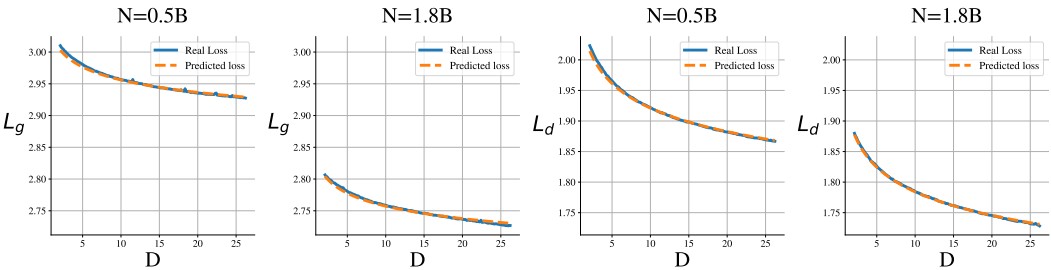

Figure 3: Effectiveness of D-CPT Law ($L_3$). (*left two*): General-corpus validation loss $L_g$ with respect to dataset size $D$ across different model sizes $N$, domain-corpus is code and general-corpus mixture ratio $r_g = 0.5$. (*right two*): Domain-corpus validation loss $L_d$ with respect to dataset size $D$ across different model sizes $N$, domain-corpus is code and domain-corpus mixture ratio $r_d = 0.5$.

$$\underset{r_d}{\operatorname{argmin}}\, L_d(N = N_0, D = D_0, r_d) \quad \text{s.t.} \quad \frac{L_g - L_g^0}{L_g^0} < T, \tag{10}$$

where $T$ is the threshold based on practical need. In Appendix G.1, given a fixed $T$, a unique optimal solution $r_d$ is obtained. To validate it in real scenarios, by applying D-CPT Law, we calculate the optimal domain-corpus mixture ratio $r_d = 0.924$ given a dataset size $D_0 = 10B$, model size $N_0 = 1.8B$, $T = 3\%$, domain-corpus is chemistry, and initial general validation loss value of $L_g^0 = 2.8602$. Table 5 presents the results of real general-corpus validation loss and domain-corpus validation loss with respect to different domain-corpus mixture ratios, we find that the real value exactly matches the predicted values($L_{g\_pred} = 2.9458$ and $L_{d\_pred} = 1.7284$), domain-corpus mixture ratio exceeding 0.924 leads to a general validation loss that surpasses the 3% threshold of $L_g^0$.

**Usage 2: Optimal mixture on limited domain-specific data**  Given that domain-corpus is typically limited relative to the abundance of the general corpus, we study how to determine the optimal mixture ratio when domain-corpus is limited and general-corpus is sufficient. Specifically, for an LLM with parameter $N_0$ and limited domain-corpus $D_d^0$, we aim to minimize the domain-corpus validation loss $L_d$ by selecting the optimal domain-corpus mixture ratio $r_d$ as follows:

$$\underset{r_d}{\operatorname{argmin}}\, L_d(N = N_0, D, r_d) \quad \text{s.t.} \quad D_d = D_d^0, \tag{11}$$

In Equation 11, we can reach the minimum value within the $0 < r_d < 1$ discussed in Appendix G.2. To validate it in real scenarios, we conducted experiments within the music domain by setting the model parameters $N_0 = 1.8B$ and the domain-specific dataset size $D_d = 5B$. As we have data points at a large scale, we fit D-CPT Law using only data where $D_d < 2B$ to align with the use case scenarios. After using the D-CPT Law, we find that the optimal domain-corpus mixture ratio is 0.732. Table 6 shows the results of real domain-corpus validation loss of the music domain. We observe that $r_d = 0.732$ yields the lowest domain-corpus validation loss. Moreover, our predicted domain-corpus validation loss is 0.7328 when $r_d = 0.732$, which is close to the real value (0.7309).

**Usage 3: Resource allocation**  D-CPT Law is consistent with Chinchilla Scaling Law under the fixed mixture ratio to address resource allocation. Specifically, how to find the optimal values of $N$ and $D$ given a fixed compute budget. Detailed results are shown in Appendix G.3.

### 4.4  Cross-Domain D-CPT Law

In Section 3.2, we have mentioned that the learnability of a specific domain is measured by DLC (i.e., $K$). For Cross-Domain D-CPT Law, $K$ must satisfy 3 core requirements: accessible, distinct, and robust. Based on these requirements, 4 different representations of $K$ are defined as follows:

$$K_1 = \frac{w_1}{k_1}, \quad K_2 = w_2 \times k_2, \quad K_3 = \frac{w_1}{k_1} + w_2 \times k_2, \quad K_4 = \frac{w_1}{k_1} + w_2 \times k_2 + \frac{w_3}{k_3}, \tag{12}$$

where $\{w_1, w_2, w_3\}$ are fitting parameters. In the approximate Taylor expansion for the validation loss function near the initial points, $\{k_1, k_2, k_3\}$ represent the first three coefficients. Due to the discrete nature of data points in practical scenarios, $\{k_1, k_2, k_3\}$ are approximated using the variants of validation loss. Specifically, $k_1$ denotes the precise value of the validation loss at the initial point, $k_2$ represents the difference in validation loss close to the initial points, and $k_3$ is an approximation of the second derivative of the validation loss near the initial points, details are provided in Appendix H. To compare these four representations of $K$, we have conducted experiments in both effectiveness and generalizability aspects.

Table 5: The real $L_g$ and $L_d$ with respect to $r_d$ in usage 1 setting

| $r_d$ | 0.9 | 0.91 | 0.92 | 0.924 | 0.93 | 0.94 | 1.0 |
|---|---|---|---|---|---|---|---|
| $L_g$ | 2.9052 | 2.9193 | 2.9376 | 2.9445 | 2.9644 | 2.9848 | 3.4667 |
| $L_d$ | 1.7321 | 1.7312 | 1.7311 | 1.7291 | 1.7279 | 1.7265 | 1.7220 |

Table 6: The real domain-corpus validation loss with respect to $r_d$ when $D_d$ is fixed with 5B.

| $r_d$ | 0.2 | 0.33 | 0.5 | 0.67 | 0.69 | 0.71 | 0.732 | 0.75 | 0.77 | 0.8 |
|---|---|---|---|---|---|---|---|---|---|---|
| $L_d$ | 0.7486 | 0.7495 | 0.7448 | 0.7402 | 0.7387 | 0.7391 | **0.7309** | 0.7339 | 0.7336 | 0.7398 |

**Effectiveness**   We utilize data points from all 6 domains for fitting and then evaluate their performance using $R^2$ and Huber loss. In Table 7, we find that 4 representations of $K$ yield comparable results in the general domain. However, $K_1$ and $K_2$ demonstrate a noticeable decline in domain-specific aspects. Although $K_4$ slightly outperforms $K_3$ in domain-specific aspects, it requires a larger number of fitting parameters. Therefore, considering the balance between fitting efficiency, fitting performance, and accessibility, We consider $K_3$ to be the optimal representation. To further visualize it, Figure 5 illustrates the predicted curves in comparison to the real curves under various settings.

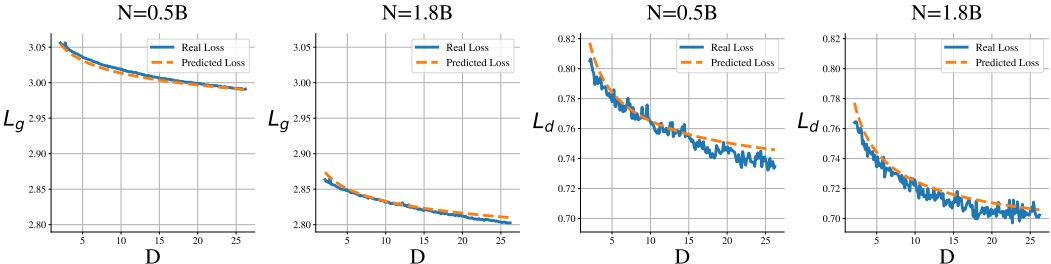

Figure 5: Effectiveness of Cross-Domain D-CPT Law ($K_3$). (*left two*): $L_g$ with respect to dataset size $D$ across different model size $N$, domain-corpus is music and $r_g$ is 0.2. (*right two*): $L_d$ with respect to dataset size $D$ across different model size $N$, domain-corpus is music and $r_d$ is 0.8.

**Generalizability**   When $K$ is identified, Cross-Domain D-CPT Law can be transformed into D-CPT Law, we believe that the former inherits the latter's generalizability in terms of model size, dataset size, and mixture ratio. In this part, we will specifically focus on discussing the domain generalizability of Cross-Domain D-CPT Law. To evaluate domain generalizability, we use data points from 4 out of 6 domains for fitting and assign the remaining 2 domains for testing. For simplicity, we only show the averaged results across 15 combinations in Table 8. Among these 4 representations of $K$, $K_3$ exhibits the superior performance, further proving its strength.

## 5 Related Works

**Scaling Law**   Many studies [26, 31, 27, 7, 16, 58] show a power-law relationship between performance and the increases in both the number of parameters and the size of the training data [31, 27, 18], which are crucial for large language models (LLMs [45, 51, 29, 3, 55, 35, 59, 36, 60]) and provide a predictive structure for determining the most efficient setups for expanded models using the insights gained from smaller models [17]. Moreover, the extension of scaling laws to autoregressive generative models widens their relevance to encompass tasks beyond text [18, 25, 17]. Recently, Muennighoff et al. [43] studied the Scaling Law of data-constrained settings by using the full pre-training dataset across multiple epochs. Ye et al. [57] investigate the data-mixing scaling law for the general LLMs to improve the pretraining efficiency.

Table 8: Domain generalizability.

| Representation | Huber loss↓ | | $R^2$ ↑ | |
|---|---|---|---|---|
| | G | D | G | D |
| $K_1$ | 0.0231 | 0.7712 | 0.9851 | 0.5855 |
| $K_2$ | 0.0222 | 2.5792 | 0.9860 | 0.5865 |
| $K_3$ | **0.0214** | **0.5335** | **0.9886** | **0.8611** |
| $K_4$ | 0.0232 | 1.1634 | 0.9849 | 0.6763 |

Table 7: The performance of 4 representations under effectiveness setting.

| Representation | Huber loss↓ | | $R^2$ ↑ | | # fitting parameters | Accessibility |
|---|---|---|---|---|---|---|
| | G | D | G | D | G/D | G/D |
| $K_1$ | 0.0675 | 0.9224 | 0.9853 | 0.9462 | + | + |
| $K_2$ | 0.0612 | 1.1924 | 0.9875 | 0.8526 | + | ++ |
| $K_3$ | **0.0566** | 0.3682 | **0.9889** | 0.9918 | ++ | ++ |
| $K_4$ | 0.0671 | **0.3396** | 0.9854 | **0.9928** | +++ | ++ |

\* For Numbers of fitting parameters, more "+" indicates a larger number of fitting parameters, implying lower fitting efficiency. Accessibility denotes the accessibility of $K$, fewer "+" signifies higher accessibility.

**Domain-specific Continual Pre-Training**  Domain-specific Continual Pre-Training aims to continually pre-train LLMs to adapt them to new domains [23, 11, 22, 30, 19]. For example, Gururangan et al. [23] introduces a growing mixture of expert architecture for domain-adaptive continual pre-training. Cossu et al. [13] show that continually pre-trained models (RoBERTa [38] and BERT [14]) are robust against catastrophic forgetting on downstream tasks. However, the above works only investigate small encoder-decoder models on limited tasks. Recently, Gupta et al. [21] study different warm-up strategies for continual pertraining for better results.

## 6    Conclusion

In this work, we have investigated the Scaling Law of Domain-specific Continual Pre-Training (D-CPT), which provides a significant step forward in the optimization of training LLMs for specific downstream domains. By developing and validating the D-CPT Law, we can easily predict the optimal mixture ratio of general and domain-specific corpora, greatly reducing the previously necessary but costly grid-searching efforts. Besides, we also adapt our D-CPT Law to the cross-domain setting and introduce the Cross-Domain D-CPT Law to further reduce the efforts of fitting the D-CPT Law of new domains. Moreover, we discuss the three practical usages of our D-CPT Law. Finally, we believe our D-CPT Law is an initial investigation into quantitative prediction methods for the domain-specific continual pre-training. With its increasing focus on data engineering, we hope our exploration facilitates further quantitative studies and theoretical analyses in this research area.

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

## A  Limitations and Future works

**Experiments on more downstream domains**  In our work, main experiments cover six downstream domains [39, 5, 24, 10, 44, 33]. In future works, it is important to experiments on more downstream domains. We will attempt to conduct experiments on CPT in more domains and fit the D-CPT Law as well as the Cross-Domain D-CPT Law.

**Experiments on more LLMs**  We primarily conduct experiments based on Qwen-1.5 and lack exploration of other Pre-Trained Base LLMs.

**Multilingualism**  We lack research on multilingualism settings. Although the medical data is in fact in Chinese, the other data are in English. Moreover, the experimental results show that the fitting results in the medical domain are poor compared to others. We lack a detailed experimental analysis of different language settings. In future research, we hope to realize cross-linguistic and multi-linguistic D-CPT Law and thereby further extend the generalizability of D-CPT Law.

**Difficulty on fitting parameters**  We find that when using L-BFGS for fitting, the initialization of the fitting parameters is essential. Different parameter initializations can lead to significantly distinct results. Besides, we find that fitting algorithms also matter, in subsequent works, we hope to compare different fitting algorithms and design methods to reduce the dependency on the initialization of the fitting parameters.

**Extensive training costs of Scaling Law**  Although we attempt to ameliorate the training costs and enhance the fitting efficiency of Scaling Law, which are detailed in Section 4.4 and Appendix I.1, Scaling Law [12, 4, 1, 37] still remains prohibitively expensive for the majority. We hope that future research endeavors will seek to reduce the training costs of Scaling Law, thereby facilitating a wider usage and understanding of these laws within the community.

## B  Broader Impacts

LLMs, particularly those involving pre-training on massive Internet data, have been identified to carry significant societal impacts and inherent biases [56, 50, 15, 2]. For instance, large language models (LLMs) may generate content that carries political bias [42]. With the rise of downstream applications of LLMs, there is a growing effort to limit their output of offensive content, rendering LLMs more controllable and mitigating their potential negative impacts. We hope that our research to make the downstream applications of LLMs more controllable.

Besides, LLMs have a significant environmental impact due to the substantial energy consumption required for their training and inference stages [46]. The extensive computational resources needed result in a high carbon footprint, thus raising concerns about the sustainability of such models in the context of global efforts to reduce greenhouse gas emissions. To this, our research can also partially reduce the consumption of GPU, thereby reducing the environmental impact of LLMs.

## C  Symbols

To enhance the reader's experience, we have listed the symbols used in this paper in Table 9.

## D  D-CPT Law with a constant Mixture Ratio

**OpenAI Scaling Law**  Kaplan et al.[31] propose a parameterization as follows:

$$L = \left[ \left( \frac{N_c}{N} \right)^{\frac{\alpha_N}{\alpha_D}} + \frac{D_c}{D} \right]^{\alpha_D}, \tag{13}$$

where $\{N_c, D_c, \alpha_N, \alpha_D\}$ are fitting parameters.

Table 9: List of symbols presented in this paper.

| Symbol | Description |
|--------|-------------|
| $r_d$ | The proportion of the domain-specific corpus within the training dataset. |
| $r_g$ | The proportion of the general corpus within the training dataset. |
| $r$ | The proportion of the target corpus within the training dataset. |
| $L_d$ | The validation loss for the domain-specific corpus. |
| $L_g$ | The validation loss for the general corpus. |
| $L$ | The validation loss for the target corpus. |
| $N$ | The size of the model parameters. |
| $D$ | The number of training tokens for the model. |
| $D_d$ | The number of training tokens of the domain-specific corpus for the model. |
| $D_g$ | The number of training tokens of the general corpus for the model. |
| $L_d^0$ | The validation loss for the domain-specific corpus before continual pre-training. |
| $L_g^0$ | The validation loss for the general corpus before continual pre-training. |

**Chinchilla Scaling Law** Continuing along the trajectory established by OpenAI Scaling Law, Hoffmann et al.[27] propose a parameterization as follows:

$$L = E + \frac{A}{N^\alpha} + \frac{B}{D^\beta}, \tag{14}$$

where $\{E, A, B, \alpha, \beta\}$ are fitting parameters. After fitting, the *Allocation* problem can be resolved by:

$$N_{opt} = G\left(\frac{C}{6}\right)^a, \quad D_{opt} = G^{-1}\left(\frac{C}{6}\right)^b, \tag{15}$$

$$\text{where} \quad G = \left(\frac{\alpha A}{\beta B}\right)^{\frac{1}{\alpha+\beta}}, \quad a = \frac{\beta}{\alpha+\beta}, \quad b = \frac{\alpha}{\alpha+\beta}, \tag{16}$$

where $N_{opt}$ and $D_{opt}$ represent the optimal value of model size and dataset size, respectively.

If we fix the mixture ratio in the training corpus, the D-CPT Law narrows down to the relationship involving only the model size $N$ and dataset size $D$. Although previous works have proposed Scaling Law to describe the relationship between variables and performance, it has not been validated under our experimental setup. Here, we present the performance of OpenAI Scaling Law and Chinchilla Scaling Law in our experimental setup. For simplicity, we present results only in the **code** domain, with a 1:1 mixture ratio. The experimental results are shown in Figure 6 and Table 10. We find that the Chinchilla Scaling Law is obviously better in our experimental setup.

Table 10: The fitting performance of two laws on code-corpus with 1:1 mixture ratio.

| **Law** | Huber loss↓ | | $R^2$ ↑ | |
|---------|-------------|---|--------|---|
| | G | D | G | D |
| OpenAI Scaling Law | 0.0026 | 0.0059 | 0.9609 | 0.8888 |
| Chinchilla Scaling Law | **0.0002** | **0.0013** | **0.9994** | **0.9925** |

# E Supplementary Materials of D-CPT Law

## E.1 Explicit trends

To provide a clear visualization of Equation 4, we have provided figures under 3 different settings, depicted in Figure 7, Figure 8, and Figure 9. All plots are trends of real data points.

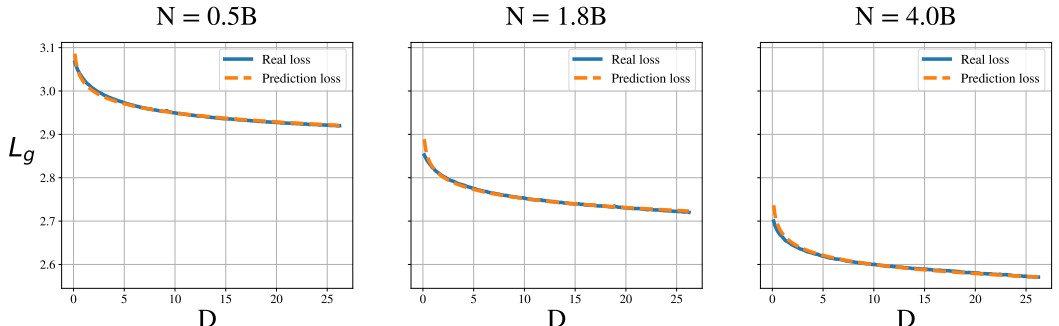

Figure 6: $L_g$ with respect to $D$ across multiple model sizes $N$. Blue solid lines stand for real data points and orange dashed lines stand for the predicted curve. Fitting law is Chinchilla Scaling Law.

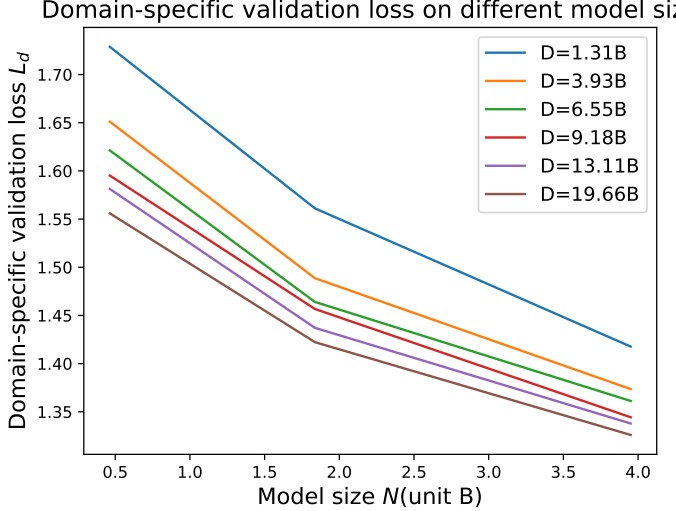

Figure 7: Domain-corpus validation loss $L_d$ with respect to model size $N$ while $\{D,r\}$ are fixed, domain-corpus is law and domain-corpus mixture ratio $r_d = 0.2$.

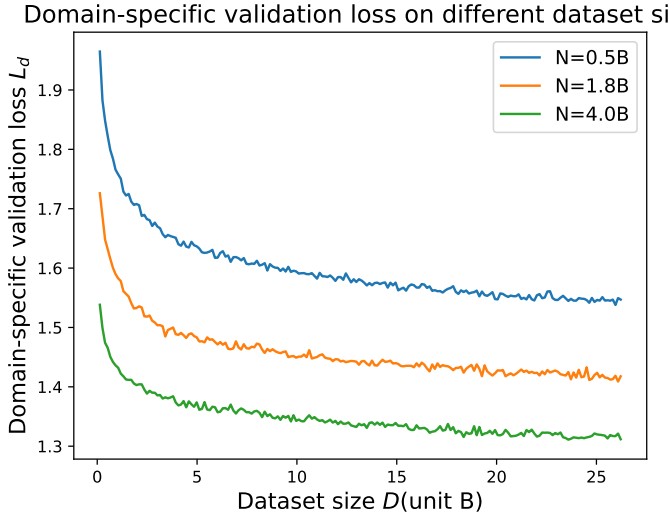

Figure 8: Domain-corpus validation loss $L_d$ with respect to dataset size $D$ while $\{N,r\}$ are fixed, domain-corpus is law and domain-corpus mixture ratio $r_d = 0.2$.

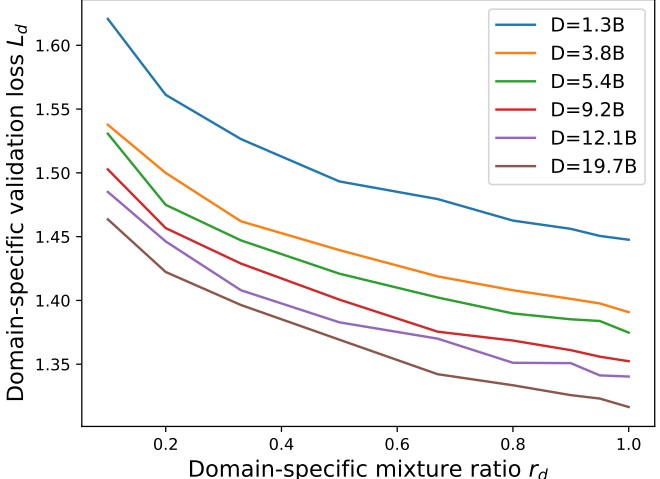

Figure 9: Domain-corpus validation loss $L_d$ with respect to domain-corpus mixture ratio $r_d$ while $\{N,D\}$ are fixed, domain-corpus is law and model size $N = 1.8B$.

## E.2 Implicit trends

In this section, we start from the perspective of experimental observations to illustrate why we can arrive at the conclusions presented in Equation 5. Subsequently, we will briefly analyze the underlying reasons for these implicit trends. For convenience, we replicate here for clarity:

$$\frac{\partial^2 L}{\partial D \partial r} < 0, \tag{17}$$

In mathematics, D-CPT Law has continuous second partial derivatives with respect to $D$ and $r$. Based on Clairaut's Theorem, we have:

$$\frac{\partial^2 L}{\partial D \partial r} = \frac{\partial^2 L}{\partial r \partial D}, \tag{18}$$

which implies that the order of partial derivative does not affect the pattern presented in Equation 17. Based on the experiments, we have plotted the approximate values of $\frac{dL_g}{dD}$ as a function of the general-corpus mixture ratio, as shown in Figure 10. Since data points are discrete, we take the difference of every 5k steps as approximate values for $\frac{dL_g}{dD}$. We present the curves of $\frac{dL_g}{dD}$ with respect to $r_g$ across multiple dataset sizes $D$. It is clear that $\frac{dL_g}{dD}$ monotonically decreases with $r_g$. Thus, based on the real experimental observations, we can infer Equation 17.

In fact, there exists an explicit relationship between $r$ and $D$, which can be represented as:

$$D_g = r_g \cdot D, \tag{19}$$
$$D_d = r_d \cdot D, \tag{20}$$
$$r_g + r_d = 1, \tag{21}$$
$$D_g + D_d = D, \tag{22}$$

where $D_g$ represents the general-corpus dataset size and $D_d$ represents the domain-corpus dataset size. If we focus on the domain-corpus validation loss, then $D_g$ is noisy data to domain-corpus, and $D_d$ is valid data to domain-corpus. If we consider $L_d$, the domain-corpus validation loss, to be solely dependent on $D_d$ and $N$, then $D$ and $r_d$ influence each other and cannot be considered independent.

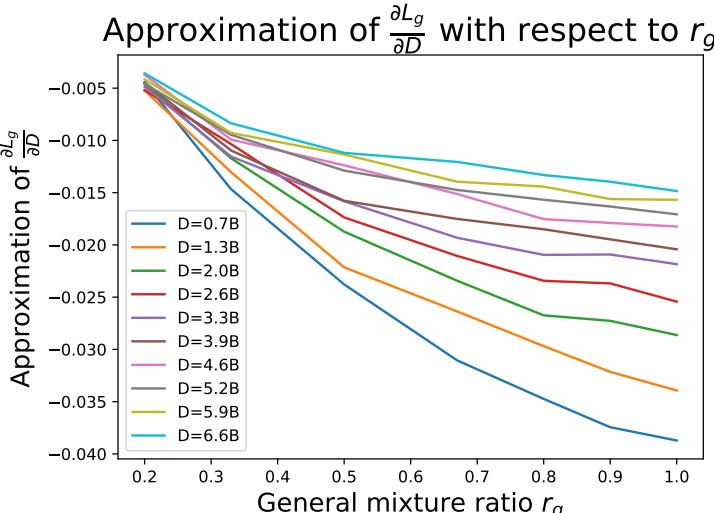

Figure 10: Approximate values of $\frac{\partial L_g}{\partial D}$ with respect to general-corpus mixture ratio $r_g$ while $\{N, D\}$ are fixed, domain-corpus is law and model size $N = 1.8B$.

Previous works have treated $N$ and $D$ as independent variables, not influencing each other. However, in our works, $D$ and $r$ are not able to be independent of each other, both from the perspective of experimental phenomena and the explicit relationship.

Additionally, we can explain Equation 5 by the principle of data efficiency. The term $\frac{dL}{dD}$ can be interpreted as the efficiency of each unit of data. With the increase of $r$, the proportion of valid data in each unit of data rises while the proportion of noisy data diminishes, resulting in enhanced efficiency of each unit of data. Given that lower loss signifies improved model performance, $\frac{dL}{dD}$ consequently displays a decreasing trend as $r$ increases.

### E.3 Details behind D-CPT Law

In this section, we will first derive and demonstrate that D-CPT law satisfies the 4 requirements mentioned in Section 3.1. Subsequently, we will briefly describe the algorithm's setup and some minor improvements.

- **Adaptability**: The newly introduced variable, the mixture ratio $r$, significantly differs from $N$ and $D$, in that the range of values for $N$ and $D$ is greater than 0, whereas $r$ is limited to the range $[0,1]$. This means that $r$ should yield valid results at both 0 and 1, and it is crucial to ensure that values of $r$ near $0^+$ or $1^-$ do not cause $L$ to exhibit infinity. The trend of $L$ with respect to $r$ generally exhibits an initially rapid and subsequently slow pattern, a behavior that can be accurately modeled by a power function. However, positioning $r$ in the denominator leads to an asymptotic increase to infinity as $r$ approaches zero from the positive direction. To mitigate this issue, we have introduced a small positive bias $\epsilon$ to $r$, which is a fitting parameter. Typically, the value of $\epsilon$ lies near 0.1. This adjustment effectively prevents explosive growth near $r = 0^+$.

- **Explicit trends**:

$$\frac{\partial L}{\partial N} = -\frac{\alpha \cdot A}{N^{\alpha+1}} < 0, \tag{23}$$

$$\frac{\partial L}{\partial D} = -\frac{\beta \cdot B \cdot r^\eta}{D^{\beta+1}} < 0, \tag{24}$$

$$\frac{\partial L}{\partial r} = \frac{B \cdot \eta}{D^\beta} \cdot r^{\eta-1} - \frac{\gamma \cdot C}{r'^{\gamma+1}}, \quad \text{where} \quad r' = r + \epsilon. \tag{25}$$

It is important to note that for the third equation, having $\frac{\partial L}{\partial r} < 0$ requires certain constraints on the fitting parameters, specifically:

$$\begin{cases} \eta > 1 \\ C > C_0 \end{cases} , \quad \text{where} \quad C_0 = \frac{B\eta\left(1+\epsilon\right)^{\gamma+1}}{\gamma D_{min}^{\beta}}. \tag{26}$$

If these two constraints are satisfied, we have:

$$\frac{\partial L}{\partial r} = \frac{B \cdot \eta}{D^{\beta}} \cdot r^{\eta-1} - \frac{\gamma \cdot C}{r'^{\gamma+1}} \tag{27}$$

$$= \frac{B\eta}{D^{\beta}r'^{\gamma+1}} \cdot \left( r^{\eta-1} r'^{\gamma+1} - \frac{\gamma C D^{\beta}}{B\eta} \right) \tag{28}$$

$$\leq \frac{B\eta}{D^{\beta}r'^{\gamma+1}} \cdot \left( (1+\epsilon)^{\gamma+1} - \frac{\gamma C D^{\beta}}{B\eta} \right) \tag{29}$$

$$\leq \frac{\gamma}{r'^{\gamma+1}} \cdot \left( \frac{B\eta\left(1+\epsilon\right)^{\gamma+1}}{D^{\beta}} - C \right) \tag{30}$$

$$\leq \frac{\gamma}{r'^{\gamma+1}} \cdot \left( \frac{B\eta\left(1+\epsilon\right)^{\gamma+1}}{D_{min}^{\beta}} - C \right) \tag{31}$$

$$\leq \frac{\gamma}{r'^{\gamma+1}} \cdot (C_0 - C) < 0. \tag{32}$$

In our experimental setup, $D$ has a minimum value, with the minimum value $D_{min}$ being approximately 0.1311B. Therefore, as long as we set $C$ greater than $C_0$ and $\eta$ greater than 1, the condition $\frac{\partial L}{\partial r} < 0$ can be satisfied. This effectively imposes constraints on the fitting parameters. In our actual fitting process, we have modified the algorithm to seamlessly incorporate these constraints. Specific details will be mentioned when introducing the algorithm.

- **Implicit trends**:

$$\frac{\partial^2 L}{\partial D \partial r} = \frac{\partial^2 L}{\partial r \partial D} = \frac{\partial \left( -\frac{\beta B r^{\eta}}{D^{\beta+1}} \right)}{\partial r} = -\frac{\eta \beta B r^{\eta-1}}{D^{\beta+1}} < 0, \tag{33}$$

- **Consistency**:

$$L(N, D, r = r_0) = E + \frac{A}{N^{\alpha}} + \frac{B \cdot r_0^{\eta}}{D^{\beta}} + \frac{C}{(r_0 + \epsilon)^{\gamma}} \tag{34}$$

$$= E_0 + \frac{A}{N^{\alpha}} + \frac{B_0}{D^{\beta}}, \tag{35}$$

$$\text{where} \quad E_0 = E + \frac{C}{(r_0 + \epsilon)^{\gamma}} \tag{36}$$

$$B_0 = B \cdot r_0^{\eta}, \tag{37}$$

which means that if $r$ is a constant $r_0$, then D-CPT Law can be transformed into a conventional Chinchilla Scaling Law. This suggests that under specific conditions where $r$ assumes a fixed value, D-CPT Law aligns with the more universally recognized Chinchilla Scaling Law.

**Constrained L-BFGS** We utilize L-BFGS to fit data points, with the objective being:

$$\min_{a,b,c,e,\alpha,\beta,\gamma,\epsilon,\eta} \text{Huber}_{\delta}(L_{fit} - \log L_{real}),$$

$$L_{fit} = \text{LSE}(e, a - \alpha \log N, b + (1 + \exp(\eta_1)) \log r - \beta \log D, c_1 - \gamma \log(r + \epsilon), c_0 - \gamma \log(r + \epsilon)),$$

$$\text{where} \quad c_0 = \log C_0, a = \log A, b = \log B, c_1 = \log C_1, e = \log E,$$

$$C = C_0 + C_1, \eta = 1 + \exp(\eta_1),$$

where LSE is the log-sum-exp operator. Our improvements to the algorithm primarily focus on the third and the last item. Previously, we mentioned that $C$ must be greater than $C_0$ to ensure the monotonic decrease of the D-CPT law with respect to $r$. Without any restrictions and fitting directly, it would sometimes lead to fitting results where $C$ does not satisfy $C \geq C_0$. Therefore, to ensure that the fitted $C$ must be greater than $C_0$, we have indirectly imposed certain restrictions on the algorithm. We decomposed the original $C$ into two parts: $C_0$ and $C_1$, and due to the characteristics of the exponential function, the fitted result of $C_1 = \exp c_1$ will be greater than 0. Consequently, $C$ will be greater than $C_0$, i.e.,

$$C = C_0 + C_1 = C_0 + \exp(c_1) > C_0, \tag{38}$$

$$\eta = 1 + \exp(\eta_1) > 1, \tag{39}$$

$$\text{where} \quad C_0 = \frac{B(1 + \exp(\eta_1))(1 + \epsilon)^{\gamma+1}}{\gamma D_{min}^{\beta}}. \tag{40}$$

Following Chinchilla Scaling Law, we find local minima of the objective function, initiating our search on a predefined grid of starting points as follows: $a \in \{-1., -0., ..., 5.\}, b \in \{-1., 0., ..., 5.\}, c \in \{-1., 0., ..., 5.\}, e \in \{-1., 0.5, ..., 1.\}, \alpha \in \{-0.5., 0., 0.5\}, \beta \in \{-0.5., 0., 0.5\}, \gamma \in \{-0.5., 0., 0.5\}, \eta_1 \in \{-0.5., 0., 0.5\}, \epsilon \in \{0., 0.5\}$. Besides, we use $\delta = 10^{-3}$ for the Huber loss.

### E.4  The choices of D-CPT Law Parameterizations

In Section 4.2, we have proposed five possible parameterizations of the D-CPT Law. After analyzing the experimental results, we have selected $L_3$ as the final parameterization of the D-CPT Law. Could there be other parameterizations better than $L_3$? We acknowledge that there may exist better parameterization than $L_3$, but we believe that this is not crucial because our core objective is to find one that satisfies the 4 requirements outlined in Section 3.1, and we assume that when these 4 requirements are met, the parameterization is considered a good option of D-CPT Law. Specifically, regarding the choice of parameterization for the D-CPT Law, our research goal can be understood as finding a parameterization that matches the trend of real (N, D, r) data points and possesses certain mathematical properties. These trends and mathematical properties can be explicitly expressed as the 4 requirements in Section 3.1: Adaptability, Explicit trends, Implicit trends, and Consistency. Besides, in Section 4.2, we provide 5 parameterizations, only $L_3$ can meet all 4 requirements.

Moreover, as there are also other parameterizations that can meet these 4 requirements, we provide another 2 parameterizations that satisfy these 4 requirements as follows:

$$L_6 = E + \frac{A}{N^\alpha} + \frac{B \cdot e^r}{D^\beta} + \frac{C}{r'^\gamma}, \tag{41}$$

$$L_7 = E + \frac{A}{N^\alpha} + \frac{B \cdot r'^\eta}{D^\beta} + \frac{C}{r'^\gamma}. \tag{42}$$

The fitting results for $L_6$, $L_7$, and $L_3$ on general-corpus and domain-corpus are as listed in Table 11. We observe that when 4 requirements are met, the fitting results do not differ a lot. In conclusion, first, as there are many parameterizations that meet the 4 requirements, it is challenging to find the optimal parameterization. Second, the $L_3$ mentioned in the paper is relatively simple and meets the 4 requirements with good fitting results.

### E.5  Compute resources

Our main experiment requires approximately 150k hours of runtime on a single A100.

Table 11: Mean performance across $L_3$, $L_6$, and $L_7$ over six domains. "G" and "D" denote general and downstream domains.

| Parameterization | Huber loss ↓ | | $R^2$ ↑ | |
|---|---|---|---|---|
| | G | D | G | D |
| $L_3$ | **0.0048** | **0.0157** | 0.9968 | 0.9796 |
| $L_6$ | 0.0051 | 0.0164 | 0.9963 | 0.9778 |
| $L_7$ | 0.0049 | 0.0160 | **0.9969** | **0.9801** |

# F  Supplementary Materials of Experiments

## F.1  Validation datasets collection

Specifically, for each domain, we first randomly select 5,000 samples from the original dataset, and then we use four open-sourced LLMs (i.e., Qwen-1.5 72B [6], Yi-34B [3], LLaMA2-13B [52], InternLM2-20B [8]) to compute the perplexity (PPL) and sort these samples based on the PPL values. Specifically, a lower PPL value denotes higher fluency of the data indicated by the model. If a sample ranks in the bottom 10% under all four open-source LLMs, we consider this sample to be noisy and exclude it. Subsequently, we randomly sample 1,000 samples from the filtered sample pool to serve as the validation set for each domain. In this way, we can obtain a high-quality validation set for all domains.

## F.2  Hyperparameters

The hyperparameters for the experiments are listed in Table 12.

Table 12: The list of hyperparameters.

| Hyperparameters | Value |
|---|---|
| Warm-up Steps | 0 |
| Gradient Accumulation Steps | 4 |
| Train Batch Size Per Device | 4 |
| Max Sequence Length | 2048 |
| Learning Rate | 3e-5 |
| Learning Rate Scheduler | cosine |
| Numbers of GPUs | 16 |

# G  Mathematical Derivation behind use case

## G.1  Usage 1

First, we will standardize the notation: $r_g$ denotes the proportion of the general corpus, $r_d$ represents the proportion of the domain-corpus, $L_g$ signifies the general-corpus validation loss, $L_d$ indicates the domain-corpus validation loss, $D$ represents the dataset size, and $N$ denotes the model size. Therefore, we have:

$$L_g = E + \frac{A}{N^\alpha} + \frac{B \cdot (1 - r_d)^\eta}{D^\beta} + \frac{C}{(1 - r_d + \epsilon)^\gamma}, \qquad (43)$$

$$L_d = E + \frac{A}{N^\alpha} + \frac{B \cdot r_d^\eta}{D^\beta} + \frac{C}{(r_d + \epsilon)^\gamma}. \qquad (44)$$

Note that we have the loss $L$ monotonically decreasing with respect to $r$, therefore we have:

$$\frac{\partial L_g}{\partial r_g} < 0 \implies \frac{\partial L_g}{\partial (1 - r_d)} < 0 \implies \frac{\partial L_g}{\partial r_d} > 0, \tag{45}$$

$$\frac{\partial L_d}{\partial r_d} < 0 \implies \frac{\partial L_d}{\partial (1 - r_g)} < 0 \implies \frac{\partial L_d}{\partial r_g} > 0, \tag{46}$$

Within the context of D-CPT, we focus on $L_d$, the domain-corpus validation loss. As the proportion of domain-corpus $r_d$ increases, $L_d$ is expected to decrease, indicating an improvement in domain-specific performance. Conversely, $L_g$, the general-corpus validation loss, is expected to increase with the growing $r_d$, suggesting a decline in general abilities. Therefore, we need to strike a balance between general and domain-specific abilities. To be specific, we will revisit the objective function of Usage 1:

$$\operatorname*{argmin}_{r_d} L_d(N = N_0, D = D_0, r_d) \quad \text{s.t.} \quad \frac{L_g - L_g^0}{L_g^0} < T, \tag{47}$$

where $L_g^0$ represents the initial general validation loss. Since $L_g$ monotonically increases with $r_d$, a maximal $r_d$ will certainly be attained under the constraint. Concurrently, as $L_d$ monotonically decreases with $r_d$, there must exist a unique $r_d$ that minimizes $L_d$.

### G.2 Usage 2

For simplicity, we restate the objective function for usage 2:

$$\operatorname*{argmin}_{r_d} L_d(N = N_0, D = \frac{D_d}{r_d}, r_d) \quad \text{s.t.} \quad D_d = D_d^0, \tag{48}$$

where $D_d$ denotes the domain-corpus dataset size, for $L_d$ in format of D-CPT Law, we have:

$$L_d(N = N_0, D = \frac{D_d}{r_d}, r_d) = E + \frac{A}{N_0^\alpha} + \frac{B r_d^\eta}{(\frac{D_d^0}{r_d})^\beta} + \frac{C}{(r_d')^\gamma}, \quad \text{where} \quad r_d' = r_d + \epsilon, \tag{49}$$

$$\frac{dL_d}{dr_d} = \frac{B(\eta + \beta)}{(D_d^0)^\beta} r_d^{\eta + \beta - 1} - \frac{\gamma C}{(r_d')^{\gamma + 1}} \implies \tag{50}$$

$$\frac{d^2 L_d}{dr_d^2} = \frac{B(\eta + \beta)(\beta + \eta - 1)}{(D_d^0)^\beta} r_d^{\eta + \beta - 2} + \frac{\gamma(\gamma + 1)C}{(r_d')^{\gamma + 2}}. \tag{51}$$

Based on Appendix E.3, we have $\eta > 1$, therefore we have:

$$\eta > 1 \implies \frac{d^2 L_d}{dr_d^2} > 0, \tag{52}$$

$$\frac{dL_d}{dr_d}(r_d = 0) = -\frac{\gamma C}{\epsilon^{\gamma + 1}} < 0, \tag{53}$$

$$\frac{dL_d}{dr_d}(r_d = 1) = \frac{B(\eta + \beta)}{(D_d^0)^\beta} - \frac{\gamma C}{(1 + \epsilon)^{\gamma + 1}}. \tag{54}$$

The derivative $\frac{dL_d}{dr_d}$ is continuously differentiable and monotonically increasing. Given that $\frac{dL_d}{dr_d}$ is negative at $r_d = 0$ and if $\frac{dL_d}{dr_d}$ is greater than 0 at $r_d = 1$, then it follows that Equation 49 attains its minimum within the interval $[0 < r_d < 1]^3$. Therefore, to ensure the existence of a valid minimum for the objective function 48, the following conditions must be satisfied:

---

[3]Solving $\frac{dL_d}{dr_d} = 0$ is relatively complex, in this works, we use MATLAB to find the roots of this function.

$$D_d^0 < \left( \frac{B(\eta + \beta)(1 + \epsilon)^\gamma + 1}{\gamma C} \right)^{\frac{1}{\beta}}. \tag{55}$$

### G.3 Usage 3

For convenience, we repeat the objective function of resource allocation as follows:

$$\underset{N,D}{\operatorname{argmin}} L(N, D) \quad \text{s.t.} \quad \text{FLOPs}(N, D) = C. \tag{56}$$

Following [31], we calculate compute budget C by:

$$C \approx 6ND. \tag{57}$$

To validate its effectiveness in real-world scenarios, we take the law domain as an example and by fixing the mixture ratio at 1:1, fit D-CPT Law. We fix compute budget $C = 5e^{19}$. Subsequently, based on the Efficient Frontier of Chinchilla[27], we obtain:

$$a = 0.6252, b = 0.3748, G = 4.1282, N_{opt} = 15.54\text{B}, D_{opt} = 0.536\text{B}. \tag{58}$$

As the closest available model size to the optimal model size indicated by Qwen1.5 is 14B, we conducted our experiments using this 14B model. The experimental results are as shown in Table 13. The experimental results reveal that the model sizes of 0.5B, 1.8B, and 4B suffer from data insufficiency. The optimal model size (14B) indeed exhibits the best performance.

Table 13: Domain-corpus validation loss with respect to various model sizes and dataset sizes while keeping the same compute budget.

| N | D | $L_d$ |
|---|---|---|
| 0.5 | 16.648 | 1.4921 |
| 1.8 | 4.588 | 1.4214 |
| 4.0 | 2.097 | 1.3552 |
| 14.0 | 0.590 | **1.3066** |

## H  Details behind Domain-specific Learnable Coefficient

In practice, the data points we obtain are discrete, thus we can only utilize approximate values to express $k_2$ and $k_3$. Specifically, we use the difference between the initial validation loss and the validation loss after 5k-steps[4] continual pre-training,i.e.,

$$k_2 = L_{0\text{steps}} - L_{5000\text{steps}}. \tag{59}$$

Besides, we define $k_3$ as the difference in the decline values between the intervals of 0 to 5k steps and 5k to 10k steps,i.e.,

$$k_3 = (L_{0\text{steps}} - L_{5000\text{steps}}) - (L_{5000\text{steps}} - L_{10000\text{steps}}) = L_{0\text{steps}} - L_{10000\text{steps}}. \tag{60}$$

Lastly, we denote $k_1$ as the validation loss obtained after training for 1k steps.

---

[4]Note that we use $L_{t\text{steps}}$ to represent the validation loss at t steps.

# I Further Analysis

## I.1 Fitting Efficiency

As each data point requires computational resources, we also investigate to improve the fitting efficiency with relatively low computational resources. In Table 14, we have compared different sampling methods for data points and introduced a decay sampling method based on the exponential decay function to enhance fitting efficiency. Specifically, we focus on the fitting efficiency across dataset size while maintaining a constant model size.

Table 14: The fitting performance of different sampling methods.

| Sampling Method | Huber loss↓ | | $R^2$ ↑ | | Resource consumption |
|---|---|---|---|---|---|
| | G | D | G | D | G/D |
| $M_1$ | **0.0041** | 0.0094 | 0.9977 | 0.9937 | 200 |
| $M_2$ | 0.0042 | 0.0103 | 0.9976 | 0.9936 | 40 |
| $M_3$ | 0.0043 | 0.0097 | 0.9978 | 0.9938 | 40 |
| $M_4$ | 0.0042 | **0.0092** | **0.9980** | **0.9941** | 45 |

\* For Resource consumption, we focus on evaluation costs and storage costs.

We have experimented with 4 different sampling methods, as follows:

- $M_1$: Dense sampling, evaluating validation loss every 1,000 steps.
- $M_2$: Sparse sampling, evaluating validation loss every 5,000 steps.
- $M_3$: Sectional sampling, evaluating every 4,000 steps in the initial 60% steps, every 8,000 steps in the remaining 40% steps.
- $M_4$: Sampling-based on an exponential decay function, detailed in Appendix I.2.

Experimental results show that the performance of $M_1$ is relatively poor. In situations where resource consumption is comparatively high, no significant improvement in fitting performance is observed, thus indicating that the sampling density in our main experiments is excessively high. The overall performance of $M_3$ and $M_4$ surpasses that of $M_2$ because both $M_3$ and $M_4$ adopt a strategy of dense sampling in the initial phase and sparser sampling in the later phase. The trend of $L$ with respect to $D$ also shifts from rapid to slow changes, and sampling more points during phases of faster decline can considerably enhance fitting efficiency. However, the sampling setup of $M_3$ is of fixed paradigm and the sampling function follows a step-wise pattern. Of course, the overall performance of $M_4$ is slightly better than $M_3$, it also offers a richer paradigm. In summary, sampling more points in the early phase of $D$ can improve the overall fitting efficiency. In practical applications, it has the potential to save on evaluation costs and storage costs.

## I.2 Decay function

In our main experiments, each experiment trains for 200,000 steps, with evaluations every 1,000 steps, resulting in a total of 200 data points. The decay function is represented as follows:

$$f(x) = e^{-\lambda x}. \tag{61}$$

For $M_4$ in Section I.1, we set the decay parameter $\lambda$ to 0.02 which yields 45 data points sampled. Figure 11 illustrates the decay function.

## I.3 Analysis of near-zero

Interestingly, we have found from the experiments that the trends between $L$ and $D$ are reversed when $r$ approaches 0, in this section, we will explore it in depth and find that D-CPT Law between $L$ and $D$ has an inflection point $r_i$ to change its trend.

We take the Law domain for example, the experimental results show that most of $L_g$ decreases strictly with $r_g$ when $N$ and $D$ are fixed, which is consistent with D-CPT Law. However, as $r_g$ approaches

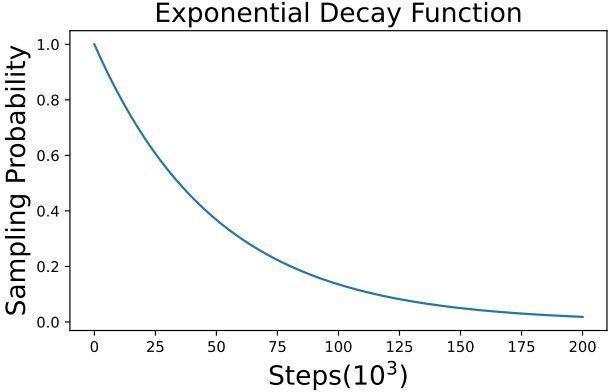

Figure 11: Illustration of decay function.

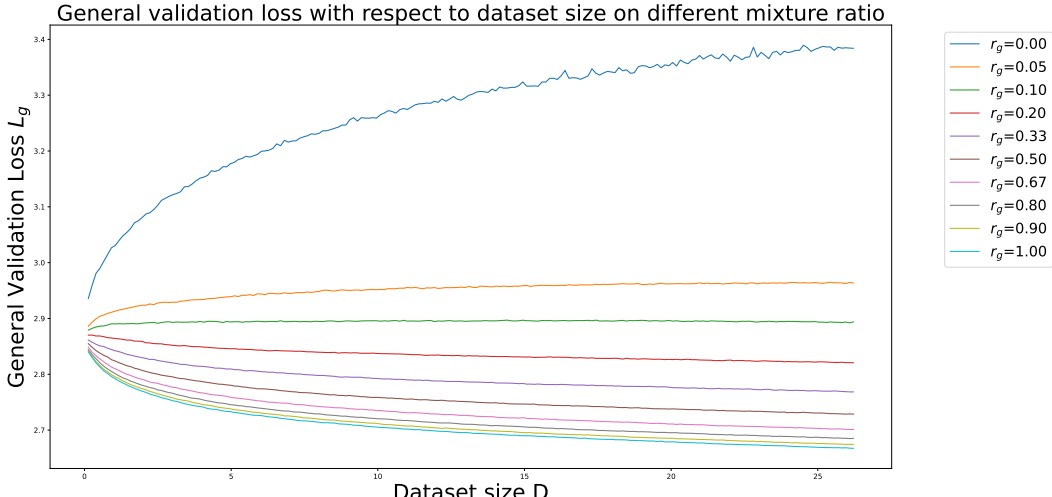

Figure 12: General validation loss with respect to dataset size across various mixture ratios, the domain-specific corpus is the law and $N = 1.8B$.

0, the trend of L changes. Through analysis of Figure 12, we observe that when $r_g$ is greater than 0.1, $L_g$ monotonically decreases with $D$, which aligns with the findings of D-CPT Law and previous works. However, when $r_g$ is less than or equal to 0.05, $L_g$ monotonically increases with $D$. This phenomenon is not limited to just one domain, we find that almost all domains exhibit this kind of behavior. We name the mixture ratio which changes the trends of $L$ as inflection point $r_i$. Accurately pinpointing $r_i$ is challenging. From an experimental perspective, it requires repeated experiments to approach $r_i$ progressively, which requires high experimental costs. Additionally, the exact value of $r_i$ changes across different domains, in our experimental setup, we find that $r_i$ for 6 domains all fall between 0 and 0.1.

When the mixture ratio is less than the inflection point, $L$ monotonically increases with $D$, which is inconsistent with the D-CPT Law. Therefore, the D-CPT Law predicts poorly when the mixture ratio is less than $r_i$. Fortunately, predictions when the mixture ratio is less than $r_i$ are meaningless in the context of our works for two reasons: (1) In practical situations, we may not be particularly concerned with cases where the mixture ratio is very small, as the inflection point in most domains is less than 0.05. (2) When the mixture ratio is lower than $r_i$, $L$ monotonically increases with $D$, meaning that as the training cost increases, the performance of the model worsens. This is contrary to our initial objective, as we hope that after D-CPT, the domain-specific ability is enhanced. Thus, predictions when the mixture ratio is less than $r_i$ are considered meaningless.

Of course, if we collect data points of small mixture ratios which means that the curves for these data points all show $L$ increasing with $D$, then we can fit these data points. In that case, the fitting parameter $B$ in D-CPT Law would be a negative value. If we know accurately the value of $r_i$, we can express D-CPT Law in form of a piecewise function or represent it with a unified equation. However, the problem lies in precisely determining the value of $r_i$. In future works, we hope to propose a low-cost method to accurately determine the value of $r_i$. For example, we could conduct experiments with both small and large mixture ratios and fit them separately, then determine the value of $r_i$ based on the intersection of two resulting laws.

# J Supplementary Tables

Table 15: Supplementary Table of Table 1. Huber loss of 5 parameterizations across 6 domains.

| Parameterization | Code | | Math | | Law | | Music | | Chemistry | | Medical | |
|---|---|---|---|---|---|---|---|---|---|---|---|---|
| | G | D | G | D | G | D | G | D | G | D | G | D |
| $L_1$ | 0.0046 | 0.0191 | 0.0049 | 0.0165 | 0.0058 | 0.0182 | 0.0033 | 0.0291 | 0.0055 | 0.0108 | 0.0141 | 0.0078 |
| $L_2$ | 0.0035 | 0.0190 | 0.0040 | 0.0165 | 0.0047 | 0.0182 | 0.0027 | 0.0275 | 0.0045 | 0.0109 | 0.0104 | 0.0077 |
| $L_3$ | 0.0036 | 0.0190 | 0.0040 | 0.0164 | 0.0046 | 0.0181 | 0.0027 | 0.0224 | 0.0044 | 0.0104 | 0.0092 | 0.0076 |
| $L_4$ | 0.0040 | 0.0195 | 0.0040 | 0.0156 | 0.0047 | 0.0183 | 0.0035 | 0.0249 | 0.0050 | 0.0096 | 0.0183 | 0.0080 |
| $L_5$ | 0.0300 | 0.0657 | 0.0302 | 0.0440 | 0.0426 | 0.0364 | 0.0188 | 0.0357 | 0.0248 | 0.0229 | 0.0501 | 0.0582 |

Table 16: Supplementary Table of Table 1. $R^2$ of 5 parameterizations across 6 domains.

| Parameterization | Code | | Math | | Law | | Music | | Chemistry | | Medical | |
|---|---|---|---|---|---|---|---|---|---|---|---|---|
| | G | D | G | D | G | D | G | D | G | D | G | D |
| $L_1$ | 0.9967 | 0.9775 | 0.9965 | 0.9911 | 0.9959 | 0.9854 | 0.9972 | 0.9596 | 0.9959 | 0.9915 | 0.9846 | 0.9551 |
| $L_2$ | 0.9977 | 0.9784 | 0.9974 | 0.9909 | 0.9971 | 0.9853 | 0.9978 | 0.9655 | 0.9970 | 0.9912 | 0.9919 | 0.9584 |
| $L_3$ | 0.9977 | 0.9783 | 0.9974 | 0.9910 | 0.9971 | 0.9853 | 0.9978 | 0.9734 | 0.9971 | 0.9915 | 0.9934 | 0.9583 |
| $L_4$ | 0.9980 | 0.9774 | 0.9980 | 0.9916 | 0.9976 | 0.9852 | 0.9970 | 0.9689 | 0.9973 | 0.9937 | 0.9735 | 0.9534 |
| $L_5$ | 0.9628 | 0.9104 | 0.9639 | 0.9665 | 0.9431 | 0.9732 | 0.9787 | 0.9542 | 0.9677 | 0.9820 | 0.8814 | 0.9208 |

Table 17: Supplementary Table of Table 2. Huber loss of 5 parameterizations across 6 domains, each unit displays the average value of 3-fold cross-validation.

| Parameterization | Code | | Math | | Law | | Music | | Chemistry | | Medical | |
|---|---|---|---|---|---|---|---|---|---|---|---|---|
| | G | D | G | D | G | D | G | D | G | D | G | D |
| $L_1$ | 0.0033 | 0.0190 | 0.0049 | 0.0170 | 0.0073 | 0.0175 | 0.0043 | 0.0202 | 0.0052 | 0.0147 | 0.0083 | 0.0145 |
| $L_2$ | 0.0048 | 0.0188 | 0.0046 | 0.0169 | 0.0049 | 0.0176 | 0.0031 | 0.0198 | 0.0049 | 0.0147 | 0.0060 | 0.0145 |
| $L_3$ | 0.0039 | 0.0185 | 0.0046 | 0.0167 | 0.0047 | 0.0176 | 0.0051 | 0.0186 | 0.0059 | 0.0144 | 0.0051 | 0.0143 |
| $L_4$ | 0.0036 | 0.0182 | 0.0036 | 0.0170 | 0.0067 | 0.0176 | 0.0054 | 0.0195 | 0.0070 | 0.0144 | 0.0064 | 0.0144 |
| $L_5$ | 0.0103 | 0.0237 | 0.0104 | 0.0157 | 0.0108 | 0.0082 | 0.0063 | 0.2523 | 0.0084 | 0.0082 | 0.0168 | 0.0389 |

Table 18: Supplementary Table of Table 2. $R^2$ of 5 parameterizations across 6 domains, each unit displays the average value of 3-fold cross-validation.

| Parameterization | Code | | Math | | Law | | Music | | Chemistry | | Medical | |
|---|---|---|---|---|---|---|---|---|---|---|---|---|
| | G | D | G | D | G | D | G | D | G | D | G | D |
| $L_1$ | 0.9583 | 0.9472 | 0.9589 | 0.9425 | 0.9549 | 0.9336 | 0.9715 | 0.9130 | 0.9582 | 0.9536 | 0.9108 | 0.9295 |
| $L_2$ | 0.9694 | 0.9536 | 0.9681 | 0.9521 | 0.9656 | 0.9301 | 0.9774 | 0.9230 | 0.9681 | 0.9529 | 0.9491 | 0.9404 |
| $L_3$ | 0.9686 | 0.9577 | 0.9672 | 0.9551 | 0.9718 | 0.9508 | 0.9811 | 0.9131 | 0.9780 | 0.9706 | 0.9598 | 0.9623 |
| $L_4$ | 0.9578 | 0.9509 | 0.9760 | 0.9535 | 0.9741 | 0.9304 | 0.9700 | 0.9293 | 0.9725 | 0.9660 | 0.9575 | 0.9419 |
| $L_5$ | 0.7411 | 0.7785 | 0.7466 | 0.5661 | 0.7008 | 0.8728 | 0.8146 | 0.9186 | 0.7158 | 0.9307 | 0.3821 | 0.8877 |

Table 19: Supplementary Table of Table 3. Huber loss of 5 parameterizations across 6 domains, each unit displays the average value of 3-fold cross-validation.

| Parameterization | Code | | Math | | Law | | Music | | Chemistry | | Medical | |
|---|---|---|---|---|---|---|---|---|---|---|---|---|
| | G | D | G | D | G | D | G | D | G | D | G | D |
| $L_1$ | 0.0029 | 0.0195 | 0.0023 | 0.0089 | 0.0027 | 0.0071 | 0.0018 | 0.0112 | 0.0032 | 0.0076 | 0.0265 | 0.0043 |
| $L_2$ | 0.0047 | 0.0252 | 0.0039 | 0.0097 | 0.0046 | 0.0072 | 0.0018 | 0.0216 | 0.0040 | 0.0055 | 0.0136 | 0.0049 |
| $L_3$ | 0.0031 | 0.0129 | 0.0030 | 0.0088 | 0.0033 | 0.0056 | 0.0019 | 0.0124 | 0.0031 | 0.0139 | 0.0059 | 0.0041 |
| $L_4$ | 0.0066 | 0.0180 | 0.0047 | 0.0093 | 0.0059 | 0.0068 | 0.0024 | 0.0121 | 0.0055 | 0.0041 | 0.0254 | 0.0054 |
| $L_5$ | 0.0120 | 0.0259 | 0.0120 | 0.0172 | 0.0123 | 0.0114 | 0.0068 | 0.0140 | 0.0093 | 0.0087 | 0.0202 | 0.0229 |

Table 20: Supplementary Table of Table 3. $R^2$ of 5 parameterizations across 6 domains, each unit displays the average value of 3-fold cross-validation.

| Parameterization | Code | | Math | | Law | | Music | | Chemistry | | Medical | |
|---|---|---|---|---|---|---|---|---|---|---|---|---|
| | G | D | G | D | G | D | G | D | G | D | G | D |
| $L_1$ | 0.9930 | 0.8001 | 0.9943 | 0.9639 | 0.9927 | 0.9827 | 0.9943 | 0.9310 | 0.9871 | 0.9093 | 0.7084 | 0.8545 |
| $L_2$ | 0.9736 | 0.7848 | 0.9760 | 0.9544 | 0.9683 | 0.9818 | 0.9939 | 0.7847 | 0.9783 | 0.9754 | 0.7212 | 0.8644 |
| $L_3$ | 0.9900 | 0.8849 | 0.9863 | 0.8435 | 0.9858 | 0.9814 | 0.9935 | 0.9014 | 0.9879 | 0.9633 | 0.9753 | 0.9012 |
| $L_4$ | 0.9453 | 0.8489 | 0.9568 | 0.9630 | 0.9296 | 0.9492 | 0.9921 | 0.8740 | 0.9468 | 0.9545 | 0.7048 | 0.8324 |
| $L_5$ | 0.8946 | 0.8309 | 0.8959 | 0.9049 | 0.8931 | 0.9142 | 0.9139 | 0.8667 | 0.9028 | 0.9173 | 0.7115 | 0.8356 |

Table 21: Supplementary Table of Table 4. Huber loss of 5 parameterizations across 6 domains, each unit displays the average value of k-fold cross-validation.

| Parameterization | Code | | Math | | Law | | Music | | Chemistry | | Medical | |
|---|---|---|---|---|---|---|---|---|---|---|---|---|
| | G | D | G | D | G | D | G | D | G | D | G | D |
| $L_1$ | 0.0014 | 0.0070 | 0.0016 | 0.0064 | 0.0020 | 0.0059 | 0.0010 | 0.0148 | 0.0018 | 0.0041 | 0.0051 | 0.0024 |
| $L_2$ | 0.0013 | 0.0077 | 0.0015 | 0.0066 | 0.0018 | 0.0059 | 0.0010 | 0.0148 | 0.0017 | 0.0042 | 0.0055 | 0.0026 |
| $L_3$ | 0.0013 | 0.0061 | 0.0015 | 0.0065 | 0.0018 | 0.0059 | 0.0010 | 0.0151 | 0.0017 | 0.0042 | 0.0041 | 0.0027 |
| $L_4$ | 0.0044 | 0.0078 | 0.0040 | 0.0074 | 0.0046 | 0.0060 | 0.0047 | 0.0104 | 0.0049 | 0.0048 | 0.0066 | 0.0038 |
| $L_5$ | 0.0067 | 0.0162 | 0.0071 | 0.0122 | 0.0090 | 0.0089 | 0.0063 | 0.0112 | 0.0087 | 0.0087 | 0.0188 | 0.0964 |

Table 22: Supplementary Table of Table 4. $R^2$ of 5 parameterizations across 6 domains, each unit displays the average value of k-fold cross-validation.

| Parameterization | Code | | Math | | Law | | Music | | Chemistry | | Medical | |
|---|---|---|---|---|---|---|---|---|---|---|---|---|
| | G | D | G | D | G | D | G | D | G | D | G | D |
| $L_1$ | 0.9978 | 0.9746 | 0.9976 | 0.9892 | 0.9965 | 0.9861 | 0.9983 | 0.9181 | 0.9971 | 0.9912 | 0.9829 | 0.9443 |
| $L_2$ | 0.9980 | 0.9719 | 0.9978 | 0.9890 | 0.9971 | 0.9861 | 0.9985 | 0.9221 | 0.9974 | 0.9911 | 0.9853 | 0.9431 |
| $L_3$ | 0.9980 | 0.9761 | 0.9977 | 0.9892 | 0.9972 | 0.9861 | 0.9984 | 0.9293 | 0.9974 | 0.9911 | 0.9899 | 0.9585 |
| $L_4$ | 0.9830 | 0.9600 | 0.9851 | 0.9844 | 0.9803 | 0.9856 | 0.9836 | 0.9404 | 0.9800 | 0.9887 | 0.9662 | 0.8886 |
| $L_5$ | 0.9801 | 0.9106 | 0.9779 | 0.9672 | 0.9670 | 0.9775 | 0.9794 | 0.9491 | 0.9674 | 0.9759 | 0.8702 | 0.2798 |

# K  Supplementary Figures

## K.1  Effectiveness of D-CPT Law

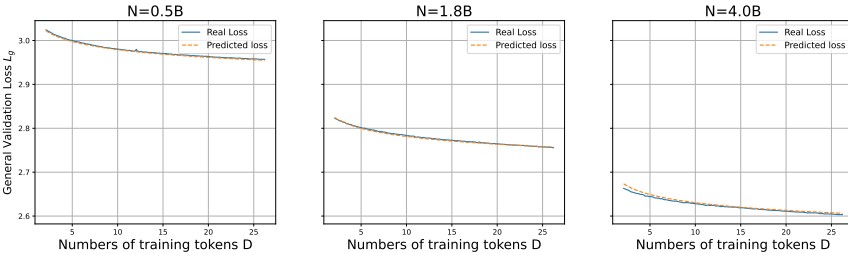

Figure 13: Effectiveness of D-CPT Law($L_3$): General-corpus validation loss $L_g$ with respect to dataset size $D$ across different model size $N$, domain-corpus is code and general-corpus mixture ratio $r_g$ is 0.33.

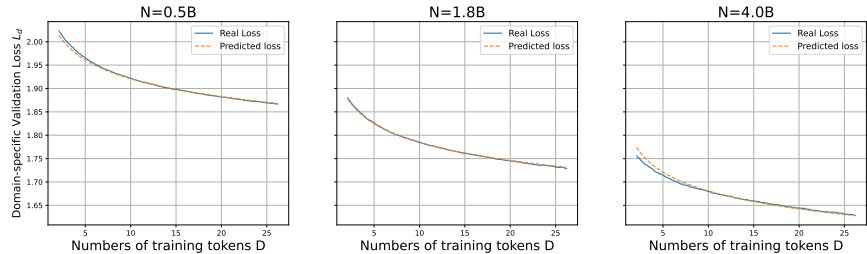

Figure 14: Effectiveness of D-CPT Law($L_3$): Domain-corpus validation loss $L_d$ with respect to dataset size $D$ across different model size $N$, domain-corpus is chemistry and domain-corpus mixture ratio $r_d$ is 0.5.

## K.2 Dataset Size Generalizability of the D-CPT Law

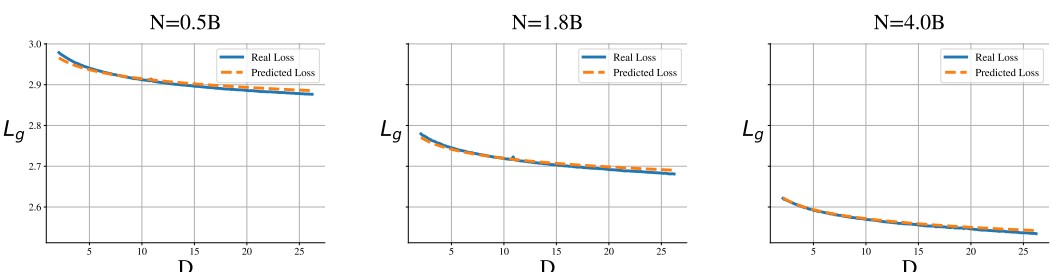

Figure 15: Dataset Size Generalizability of the D-CPT Law: General-corpus validation loss $L_g$ with respect to dataset size $D$ across various model sizes $N$, domain-corpus is math and general-corpus mixture ratio $r_g = 0.8$. The experiments use data from the first 2/3 of the steps for fitting, to verify whether the D-CPT Law exhibits generalizability across different dataset sizes.

## K.3 Domain Generalizability of the Cross-Domain D-CPT Law

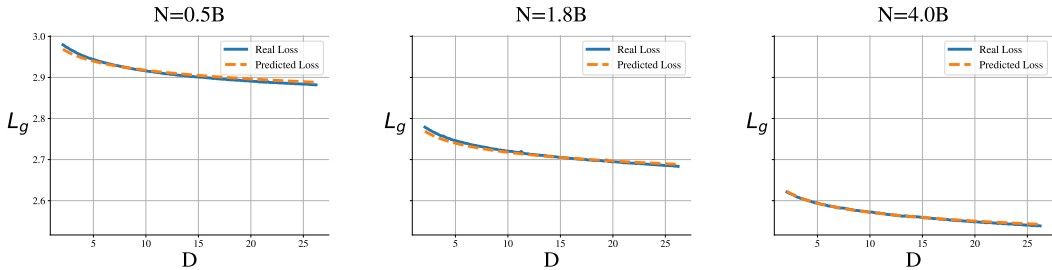

Figure 16: Domain Generalizability of the Cross-Domain D-CPT Law: General-corpus validation loss $L_g$ with respect to dataset size $D$ across various model sizes $N$, domain-corpus is Music and general-corpus mixture ratio $r_g = 0.8$. The experiments use data points from {Code, Math, Law, Medical} domains for fitting, to verify whether the Cross D-CPT Law exhibits generalizability across different domains.

Figure 17: Domain Generalizability of the Cross-Domain D-CPT Law: General-corpus validation loss $L_g$ with respect to dataset size $D$ across various model sizes $N$, domain-corpus is Chemistry and general-corpus mixture ratio $r_g = 0.8$. The experiments use data points from {Code, Math, Law, Medical} domains for fitting, to verify whether the Cross D-CPT Law exhibits generalizability across different domains.

