# OpenReview forum: "D-CPT Law: Domain-specific Continual Pre-Training Scaling Law for Large Language Models"
_NeurIPS.cc/2024/Conference — NeurIPS 2024 poster_

### Official Review · Reviewer_sFHX · 2024-07-11

**Soundness:** 2
**Presentation:** 3
**Contribution:** 3
**Rating:** 5
**Confidence:** 3

**Summary:**

This work proposes a domain-specific scaling law to optimize mixture ratio between the general domain training corpus and the specific domain corpus. The authors considered several parameterizations and experimented on six different downstream domains such as code, math, and Medical. The result show that D-CPT Law has a high fitting accuracy across various model sizes, dataset sizes, and mixture ratios. Furthermore, the proposed cross-domain D-CPT Law reduced training costs while maintaining high performance in predicting domain-specific continual pre-training results.

**Strengths:**

1. The paper proposes a domain-specific scaling that can more efficiently determine thec optimal mixture ratio between general and domain-specific corpora. The experiments demonstrate its effectiveness.

2. The paper presents three applications of D-CPT Law that may be of interest and benefit to the community.

**Weaknesses:**

1. The models are limited to the Qwen series, which are relatively small in size. This makes it unclear whether similar results would be observed with larger models.

**Questions:**

1. The author stated, "we test the validation loss every 1,000 steps and the total training steps are 20k." How did you determine the step parameters, such as 1,000 and 20,000?

2. You defined five parameterizations and demonstrated that L3 is the best through experiments. Could there be other parameterizations better than L3? In other words, is it possible to theoretically prove that L3 is the optimal D-CPT Law?

**Limitations:**

The authors state several limitations and will consider addressing them in their future work.

---

> ### Author Rebuttal · Authors · 2024-08-07
>
> Thank you very much for your valuable comments. I appreciate the time and effort you have put into reviewing our manuscript. Below, I will address your concerns and provide further clarifications.
>
> **Weakness**
>
> We acknowledge that the current status of our works is limited to the Qwen series (as we have also mentioned in the limitations) and the size of our model is relatively small (0.5 B to 4 B parameters). However, I will explain why we chose the Qwen series and the reasons behind the model size from the following points:
> 1. **The Choice of the Qwen Series**: Our work focuses on domain-specific continual pretraining(D-CPT), thus we need open-source base LLMs with strong foundational capabilities. Additionally, as we aim to explore performance across different model sizes, it is necessary that the model series have multiple model sizes. To meet these requirements, we choose the Qwen-1.5 series as this series provides multiple model versions (0.5B,1.8B,4B,7B,14B,32B,72B,110B). In contrast,  LLaMA2/3 only has 3 available model sizes (7B,13B,70B),  and the training costs on these models are large. While Pythia also offers a variety of model sizes, its overall model performance is relatively poor, making it unsuitable for D-CPT in practical scenarios.
> 2. **Relevantly Large Model Size**: Regarding the comparatively smaller model sizes mentioned, we believe our model sizes and dataset sizes are already quite large (model sizes: 0.5B,1.8B,4B, and dataset sizes: 0.1B-26B), especially when compared to 2 related works about data mixture (Data Mixing Law(https://arxiv.org/pdf/2403.16952): model sizes: fit on 70M, 160M and validate on 1B, dataset sizes: 30B; Regmix(https://arxiv.org/pdf/2407.01492): model sizes: fit on 1M and validate on 1B, dataset sizes: 1B). Additionally, we have also conducted experiments on 7B model to demonstrate that similar results will be observed with larger models (see Section 4.2, line 183-line 189). Besides, our primary focus is on the data mixture, and increasing the model size would result in a huge rise in experimental costs. Therefore, we choose these model sizes as the training and evaluation computation costs are relatively acceptable.
>
> Lastly,  the recent work Observational Scaling Law (https://arxiv.org/pdf/2405.10938) has explored the Scaling Law across different models, they stated that "model families only vary in their efficiency in converting training compute to capabilities." We agree with their perspective and we think the overall experimental patterns and findings observed in the Qwen series can also be extended to other model families.
>
> **Q1**
>
> In the Introduction section (line 50), we mention "dataset sizes from 0.1B to 26B tokens". However, there is a typo in Section 4.1 (line 159) on the training setup: "20k" should be "200k." Thank you for your question, which helped me discover this typo.
>
> Then, I would like to explain the conversion between the training step (t) and dataset size (D). The global batch size is 64, and the length of single data is 2048. The unit of dataset size is B, so the relationship between them is:
> $$D=t\cdot64\cdot2048\cdot10^{-9}$$
> The total training step is 200k, which corresponds to a dataset size of 26.21 B, while 1,000 training steps correspond to a dataset size of 0.131 B.
>
> For the validation step count of 1000, we analyzed the fitting efficiency of different data sampling methods in Appendix I (line 621- line 643), which includes results for the validation step count of 5000. We conduct supplementary experiments with different validation steps on general-corpus validation loss. The fitting results (validation data points under different settings all correspond to the data points of the 1000 validation steps) are as follows:
> |Validation Step|Huber loss|$R^2$|
> |-|-|-|
> |1000|0.0041|0.9977|
> |2000|0.0043|0.9970|
> |3000|0.0045|0.9968|
> |4000|0.0042|0.9973|
> |5000|0.0042|0.9976|
> |10000|0.0053|0.9953|
>
> We observe that the results are quite similar across different validation steps, indicating that the final findings of D-CPT Law remain consistent regardless of the chosen validation step. Moreover, selecting 1000 as the validation step ensures a sufficient number of data points.
>
> The total training steps are set to 200k based on 3 main considerations:
> 1. Our experimental setup assumes that all data is not repeated (each data is trained only once). While the general-corpus Dolma is sufficient, the domain-corpus might be insufficient. We have 6 domains, the total training steps must be less than the amount of domain-corpus. Setting the total training steps to 200k ensures no repetition of domain-corpus.
> 2. Our total experimental cost is 5.3071 e+10 TFLOPs (compared to 9.864 e+10 TFLOPs for the Data-Constrained Scaling Law(https://arxiv.org/pdf/2305.16264)). Given the already high cost, increasing the total training steps would largely increase the experimental cost. Therefore, we chose 200k steps to maintain a reasonable experimental cost.
> 3. We observed that the validation loss stabilizes around 100k-150k, so increasing the validation steps further will not lead to significant changes in the validation loss. Here, I will provide some actual data to support this point. In the music domain, for a 1.8B model, using a 1:1 data ratio, the following table shows the general-corpus validation loss and domain-corpus validation loss at different training steps.
>
> |Training Steps|General-corpus validation loss| Domain-corpus validation loss|
> |-|-|-|
> |10000|2.8191|0.8005|
> |50000|2.7674|0.7526|
> |60000|2.7612|0.7462|
> |70000|2.7556|0.7460|
> |80000|2.7505|0.7447|
> |90000|2.7471|0.7501|
> |100000|2.7432|0.7398|
> |120000|2.7367|0.7338|
> |140000|2.7312|0.7350|
> |160000|2.7253|0.7301|
> |180000|2.7206|0.7298|
> |200000|2.7159|0.7270|
>
> We can observe that the changes in the model after 100k are not as significant as those before 100k and the loss is relatively stable when the training step is close to 200k.
>
> **Q2**
>
> Please see **G.Q1** in Author Rebuttal.

---

> > ### Author Response · Authors · 2024-08-12
> > **Looking forward to Feedback as Discussion Deadline Approaches**
> >
> > Hi, we sincerely thank you very much for these constructive comments and evaluation of our manuscript. As the discussion phase will be closed on Aug 13 11:59 pm AoE, we would like to kindly ask you to take a look at our responses and reevaluate our work based on our clarifications. Please let us know whether our response addresses your concerns or whether there is any further detail we can provide to help address these concerns.
> >
> > Thank you again for dedicating your time to reviewing our paper.

---

> > ### Comment · Reviewer_sFHX · 2024-08-13
> >
> > Thank you for your responses. After reviewing the rebuttal, I will maintain the previous score.

---

### Official Review · Reviewer_rpUb · 2024-07-11

**Soundness:** 3
**Presentation:** 3
**Contribution:** 3
**Rating:** 6
**Confidence:** 5

**Summary:**

This work explore the data mixture scaling law ( D-CPT Law) between the general corpus and the downstream domain-corpus for the continual pre-training of large language model.They also extend the fitted standard D-CPT Law on cross-domain settings and propose the Cross-Domain D-CPT Law to  predict the D-CPT law of target domains.  D-CPT Law could be applied on three important scenarios: optimal mixture on the trade-off between general and domain-specific abilities, optimal  mixture for limited domain-specific data, and resource allocation setting, which is useful in practice.

**Strengths:**

1.This work experimented and analyzed on three scenarios including optimal mixture on the trade-off between general and domain-specific abilities, optimal  mixture for limited domain-specific data, and resource allocation setting, which is useful in practice
2. D-CPT Law considered both the in-domain setting and the cross-domain setting with a new involved parameter K, which provide a reference for the unseen domain data mixing.
3. The analysis on the selected fitting function is comprehensive and make sense.

**Weaknesses:**

This work focus on the data mixing scaling law for the continue pre-training phase of LLM. However, I think this work didn’t specifically have any design for the continue pre-training. The setting for the final scaling Function (Equation 6) have not mentioned any specific for continue pre-training. As the author mentioned , Equation 6  is supposed to transform into the Chinchilla  Scaling Law (which is for a LLM training from scratch) when r is fixed.

**Questions:**

1.Could you explain why this scaling law function is for continue pre-training?
2.In Equation 7, the newly involved parameter K, why is also in a similar power law pattern? The domain-specific learnable coefficient might be a much different dimension with the total number of model parameters and training data.
3.How many combination did you choose for the experiments? There are many between the data sizes from 0.1B to 26B. The way the portfolio is chosen has an impact on the on the average results.

**Limitations:**

As the authors have stated comprehensive in the limitation Section, I have no further comments for this part.

---

> ### Author Rebuttal · Authors · 2024-08-07
>
> Thanks for your careful reading and constructive suggestions. We will address your concerns shown below in detail.
>
> **Weakness**
>
> From the view of validation loss, continual pre-training (CPT) and pre-training (PT) are consistent, with the main difference being the initial validation loss (CPT: <5, PT: >10). This difference is represented by the bias term E in the D-CPT Law, and the fitted E values will differ between CPT and PT.
>
> In addition, the scenario of our work is Domain-specific Continual Pre-Training (D-CPT). In this setting, we usually need to collect high-quality domain-corpus to enhance the downstream performance and general-corpus to mitigate catastrophic forgetting of the general abilities. There, the mixture ratio (i.e., r) between domain-corpus and general-corpus plays an important role in the D-CPT setting, which is also the key distinction from the Chinchilla Scaling Law. Besides, though r is also introduced in PT, it's typically multidimensional (e.g., different data sources in CommonCrawl). In the D-CPT setting, r is two-dimensional.
>
> Moreover, we think that the consistency between the scaling laws in CPT and PT offers several benefits. First, it enables a seamless connection between PT and CPT. Specifically, by using a step function or other methods, we can jointly represent the scaling laws of both phases in a uniform parameterization. Second, when r is fixed and the D-CPT Law becomes equivalent to the Chinchilla Scaling Law, it ensures that the D-CPT Law can address resource allocation issues (line 235-line 237).
>
> Therefore, we designed such D-CPT Law. In the future, we will continue to investigate how to unify the Scaling Law for both PT and CPT.
>
> **Q1**
>
> The Scaling Law is derived from fitting data points obtained from real experiments. Therefore, a good parameterization for D-CPT Law must align with the trends observed in actual data, only then can the parameterization effectively fit the data. In Section 3.1 (line 89-line 117), we listed 4 requirements that a good parameterization should meet: Adaptability, Explicit Trends, Implicit Trends, and Consistency. Both Explicit Trends and Implicit Trends refer to the observable trends in actual D-CPT scenarios. The final form of the D-CPT Law (Equation 6) successfully meets these 4 requirements (detailed mathematical derivation is provided in line 521-line 564). We believe that as long as these 4 requirements are met, the parameterization is sufficiently robust. For more information, please see **G.Q1** in Author Rebuttal.
>
> **Q2**
>
> In Section 3.2, I mentioned K, which represents the learnability of a domain. Unlike N, D, and r, which are real variables, K is an abstract variable for which we wish to establish certain predefined properties, such as Uniformity and Monotonicity, as discussed in Section 3.2 (line 130-line 135). We incorporate this variable into the D-CPT Law using a power-law form (both in OpenAI Scaling Law(https://arxiv.org/pdf/2001.08361) and Chinchilla Scaling Law(https://arxiv.org/pdf/2203.15556), they use a power-law form to represent the scaling law). Our main objective is to first **find a method to represent K (i.e., to model K)**, and then we focus on the form of K in the D-CPT Law, where we directly use the power-law form to ensure it meets the requirements of Uniformity and Monotonicity.
>
> Secondly, the dimensions of K differ from those of N and D. This difference is not only reflected in their value ranges (N: 0.5 to 4, while K: 0.5 to 1) but also in the fitting parameters of their respective power-law forms. For instance, the fitting parameters for N, denoted as A=4.18 and $\alpha$=1.61. In contrast, the fitting parameters for K, denoted as F=0.35 and $\mu$=0.55.
>
> Finally, we added a comparison of the fitting results with different K forms. I supplemented 2 parameterizations:
> $$L_2=E+\frac{A}{N^\alpha}+\frac{Br^\eta}{D^\beta}+\frac{C}{r'^\gamma}+\frac{F}{\mu^K}$$
> $$L_3=E+\frac{A}{N^\alpha}+\frac{Br^\eta}{D^\beta}+\frac{C}{r'^\gamma}+\frac{F}{(-\log K)^\mu}$$
> The fitting results for L1(original form), L2, and L3 on general-corpus are as follows:
> |Paramterization|Huber loss|$R^2$|
> |-|-|-|
> |L1|**0.0214**|**0.9886**|
> |L2|0.0256|0.9781|
> |L3|0.0296|0.9847|
>
> It can be observed that the fitting results for different parameterizations do not show significant differences, so we believe it is more important to model the representation form of K.
>
> Additionally, modeling K with the loss (equation 9) is very intuitive:
> $$K=\frac{w_1}{k_1}+w_2\times k_2,$$
> where k1 represents the initial loss of the model. A higher value indicates the model's weaker ability in this domain, making it harder to learn, so it is inversely proportional. k2 represents the rate of decline in the loss for this domain. A higher value indicates better mastery of this domain, so it is directly proportional. We use the loss for modeling K because loss is a very direct assessment of model performance (also used in Compression(https://arxiv.org/pdf/2404.09937) to represent compression).
>
> **Q3**
>
> In the main experiment (Section 4.2), there are 6 domains, 3 model sizes, 200 dataset sizes, and 9 data mixtures.
> - **Effectiveness**:For the Effectiveness experiment, we fit and evaluate using all the data points.
> - **Model Size Generalizability**: we use 3-fold cross-validation. We randomly select 2 model sizes for fitting and then use the fitted D-CPT Law to predict the remaining model size. This yields 3 cross-validation experiments, and we take the average result.
> - **Dataset Size Generalizability**: we divide the 200 dataset size data points into 3 groups (1-67, 66-133, 133-200). Using 3-fold cross-validation, we fit two groups and validate the remaining group. This yields 3 cross-validation experiments, and we take the average result.
> - **Mixture Ratio Generalizability**: we randomly select 7 ratios for fitting and use the remaining 2 ratios for validation. With 36 combinations of selecting 7 out of 9 ratios, we take the average.

---

> ### Comment · Reviewer_rpUb · 2024-08-12
>
> I thoroughly looked at the author's response, and the score remained unchanged.

---

> > ### Author Response · Authors · 2024-08-12
> >
> > Thanks for your feedback. We will carefully address your concerns in our new version.

---

### Official Review · Reviewer_ycBX · 2024-07-14

**Soundness:** 3
**Presentation:** 3
**Contribution:** 3
**Rating:** 7
**Confidence:** 4

**Summary:**

For continual pretraining of domain-specific large language models, an important question is how to choose the optimal mixture ratio between the general corpus and the downstream domain corpus. This paper proposes to fit the scaling law for domain-specific continual pre-training (D-CPT Law), using small-scale training costs to predict the general and downstream performance of arbitrary model sizes, data sizes, and data mixture ratios. They also extend it to Cross-Domain D-CPT Law to reduce training costs for the target domain. The paper shows empirical evidence for the effectiveness and applications of the Laws.

**Strengths:**

- Discovered that model performance can be predicted given the data mixture ratio between general and domain-specific corpus and a few other factors in domain-specific language model training.
- Proposed essential requirements of the scaling law formula, compared multiple parametrization of the scaling law, and proposed a parametrization that extends the Chinchilla Scaling Law.
- Extended the law to cross-domain scenarios and described its features.
- Empirically verified the effectiveness, model size generalizability, and dataset size generalizability of the proposed methods.

**Weaknesses:**

- The paper lacks a comparison between applying the proposed laws and performing a grid search over hyperparameters. How much computation is saved in various applications?

**Questions:**

- Is it possible to fit a set of parameters in the scaling law, such that all independent variables can change their values?
- How many different $(N, D, r)$ are needed to fit the parameters in the D-CPT Law formula?

**Limitations:**

The authors addressed the limitations well in Appendix A.

---

> ### Author Rebuttal · Authors · 2024-08-07
>
> Thank you very much for your valuable comments and questions. I appreciate the time and effort you have put into reviewing our manuscript. Below, I address your concerns and provide further clarifications.
>
> **Weakness**
>
> First, the main motivation of our paper is to address the challenge of determining the data mixture ratio between general-corpus and domain-corpus during domain-specific continual pre-training. Specifically, we aim to predict performance for any given N, D, and r using scaling laws. To achieve this, we conduct multiple experiments to collect data points for fitting the D-CPT Law and then use the L-BFGS algorithm to fit the parameters of the D-CPT Law. Once we have fitted the D-CPT Law for a specific domain, we can use it to predict model performance for different N, D, and r settings. Therefore, our D-CPT Law can be considered as an **auxiliary tool**, which can be used to estimate validation loss in various scenarios and real-world usages once we obtain the D-CPT Law. In contrast, grid search typically addresses a single search problem under one setting, and the results of grid search are usually **not reusable** for other settings. Therefore, it would be **unfair** to directly compare the experimental costs of the two methods based on these aspects. However, for clarity, we will show the differences in experimental costs between using the D-CPT Law and grid search for a specific setting, as illustrated in Table 5 of the paper.
>
> **Cost of D-CPT Law**: We take the setting of Table 5 as an example. Specifically, the model size is 1.8 B, and the dataset size is 10B (See Line 214-Line 221). We have prepared data points for 9 different ratios, so the total compute budget required for fitting D-CPT Law is:
> $$1.8 * 10^9 * 10 * 10^9 * 9 * 6/10^{12}=9.72*10^8\  \text{TFLOPs}$$
> **Cost of Grid Search**: In Table 5, both the D-CPT Law and actual experimental data indicate that the optimal data ratio is 0.924. To perform a grid search on this ratio, based on binary search, we start by conducting experiments with r=0 and r=1. Then we would test r=0.5, followed by r=0.75, r=0.875, r=0.9375, r=0.90625, r=0.921875, r=0.9296875, r=0.92578125, and r=0.923828125. Assuming this meets the requirements under the settings of Table 5, a total of 11 experiments on different ratios would be needed. The total compute budget is:
> $$1.8 * 10^9 * 10 * 10^9 * 11 * 6 /10^{12}=11.88 * 10^8 \ \text{TFLOPs}$$
> By comparison, it can be observed that in the single setting of Table 5, the experimental cost of grid search is still higher than that of D-CPT Law. Moreover, a significant feature of scaling laws is that the performance of small-scale experiments can be used to predict the performance of large-scale experiments. Therefore, for a 14B model, D-CPT Law can be experimented on at 1.8B, whereas grid search needs to be experimented on at 14B. This results in a tenfold difference in experimental cost for a single experiment, and the results of the grid search cannot be **reused**. Furthermore, if the grid search needs to evaluate 10 experimental settings, the difference would grow to 100 times (14 / 1.8 *11.88 / 9.72 * 10 = 95.06 times). Therefore, the D-CPT Law significantly reduces the experimental cost required by grid search, and it can be reused across multiple settings.
>
> We appreciate your suggestion and will include this supplementary information in our new version to visually illustrate the cost difference between grid search and D-CPT Law.
>
> **Q1**
>
> For clarity, here I first briefly introduce the D-CPT Law formulation and the fitting process as follows:
>
> $$L(N,D,r)=E+\frac{A}{N^\alpha}+\frac{B\cdot r^\eta}{D^\beta}+\frac{C}{(r+\epsilon)^\gamma},$$
> where N, D, and r are independent variables and the other parameters as fitting parameters. After collecting multiple data points for (N, D, r), the L-BFGS algorithm is used to fit the D-CPT Law and obtain specific values for the fitting parameters E, A, B, C, $\epsilon$, $\gamma\$, $\alpha$, and $\beta$. Then, we can use the D-CPT Law to predict L for any given N, D, and r.
>
> For your question, I have **2 interpretations**. I will explain each situation separately:
> 1. If you mean fitting the relationships between L and N, D, and r separately, as done in the OpenAI Scaling Law(https://arxiv.org/pdf/2001.08361),  I have added the fitting formulation for L versus D and L versus r as follows:
> $$ L(D) = E_D + \frac{B}{D^\beta}, \quad \text{when N, r are fixed} $$
> $$ L(r) = E_r + \frac{C}{r^\gamma}, \quad \text{when N,D are fixed} $$
> Similarly, the fitting parameters of the above formula can be obtained using L-BFGS (for instance, we have fitted the above formula and we get B=2.2075, $E_D$=0.6175585, $\beta$=0.014580619, C=0.49847686, $E_r$=2.1966128, and $\gamma$=0.13764255). This allows us to individually explore the relationship between L and D for a given N and r, as well as the relationship between L and r for a given N and D. However, the goal of our work is to explore the relationship between L and (N, D,r) together. The relationships L(D) and L(r) are insufficient for exploring the behavior of L with respect to arbitrary values of N, D, and r.
> 2. If you mean fitting the relationship among L and (N, D, r) (as our paper does), each independent variable can change its values independently. However, in terms of the function's trend, D and r are interrelated (see line 106 - line 110 ). Specifically, changes in r will affect the trend of L with respect to D, and changes in D will affect the trend of L with respect to r. This can be seen from the form of Equation 5, and further explanation can be found in Appendix E.2 of the paper. Additionally, when N and D are fixed, L only depends on r; when N and r are fixed, L only depends on D; and when D and r are fixed, L only depends on N. This leads to the above situation where we fit L(D) and L(r) independently.
>
> **Q2**
>
> Please see **G.Q2** in Author Rebuttal.

---

> > ### Author Response · Authors · 2024-08-12
> > **Looking forward to Feedback as Discussion Deadline Approaches**
> >
> > Hi, we sincerely thank you very much for these constructive comments and evaluation of our manuscript. As the discussion phase will be closed on Aug 13 11:59 pm AoE, we would like to kindly ask you to take a look at our responses and reevaluate our work based on our clarifications. Please let us know whether our response addresses your concerns or whether there is any further detail we can provide to help address these concerns.
> >
> > Thank you again for dedicating your time to reviewing our paper.

---

> > > ### Comment · Reviewer_ycBX · 2024-08-14
> > >
> > > Thank you for your responses. My questions are addressed and I have raised my rating.

---

### Author Rebuttal · Authors · 2024-08-07

# General Response
Thanks a lot for handling/reviewing our submitted manuscript. We would like to thank the reviewers for their thoughtful and constructive comments and suggestions. By addressing each of the issues raised by the reviewers, we believe that the quality and clarity of our D-CPT Law can be improved a lot. The general responses are summarized as follows:

**G.Q1:  Could there be other parameterizations better than L3?**

We acknowledge that there may exist better parameterization than L3, but we believe that this is not crucial because our core objective is to find one that satisfies the 4 requirements outlined in Section 3.1 (Line 100-Line 110), and we assume that when these 4 requirements are met, the parameterization is considered a good option of D-CPT Law. Specifically, regarding the choice of parameterization for the D-CPT Law, our research goal can be understood as finding a parameterization that matches the trend of real (N, D, r) data points and possesses certain mathematical properties. These trends and mathematical properties can be explicitly expressed as the 4 requirements in Section 3.1: Adaptability, Explicit trends, Implicit trends, and Consistency. Besides, in Section 4.2, we provide 5 parameterizations, only L3 can meet all 4 requirements.

Moreover, as there are also other parameterizations that can meet these 4 requirements, we provide another 2 parameterizations that satisfy these 4 requirements as follows:

$$ L_6 = E + \frac{A}{N^{\alpha}} + \frac{B \cdot e^{r}}{D^\beta} + \frac{C}{r'^{\gamma}} $$

$$L_7 = E + \frac{A}{N^{\alpha}} + \frac{B \cdot r'^\eta}{D^\beta} + \frac{C}{r'^{\gamma}} $$

The fitting results for L6, L7, and L3 on general-corpus are as follows:
| Paramterization | Huber loss | $R^2$ |
| ---------------- | ---------- | ----- |
| $L_3$              | 0.0048     | 0.9968  |
| $L_6$             | 0.0051     | 0.9963  |
| $L_7$             | 0.0049     | 0.9969  |

The fitting results for L6, L7, and L3 on domain-corpus are as follows:

| Paramterization | Huber loss | $R^2$ |
| ---------------- | ---------- | ----- |
| $L_3$              | 0.0157     | 0.9796  |
| $L_6$             | 0.0164     | 0.9778  |
| $L_7$             | 0.0160     | 0.9801 |

We observe that when 4 requirements are met, the fitting results do not  differ a lot.
In conclusion, first, as there are many parameterizations that meet the 4 requirements, it is challenging to  find the optimal parameterization. Second, the L3 mentioned in the paper is relatively simple and meets the 4 requirements with good fitting results. In the future, we will continue to investigate how to find the optimal parameterization for the D-CPT Law.

**G.Q2: How many different (N, D, r) are needed to fit the parameters in the D-CPT Law formula?**

In the main experiment, we have a total of 6 domains, 3 model sizes (0.5B, 1.8B, 4B), 9 data mixtures, and 200 validation loss data points (collected every 1k training steps for a total of 200k training steps). Hence, the main experiment consists of 6x3x9x200 = 32,400 data points. In other words, there are 5,400 data points for each domain.

Additionally, in Appendix I (see line 621 - line 643), we analyzed the impact of data point sampling methods on the fitting results. We found that sampling more points when D is relatively small, and sampling fewer points when D is relatively large, can greatly reduce the experimental costs.

Besides, we found that validating every 5000 steps and every 1000 steps resulted in similar fitting results. Therefore, we can actually achieve a good fitting result with only 1/5 of the original data points, which means we only need 5400/5=1080 data points to fit. Therefore, in practical scenarios, considering fitting efficiency, approximately **1000** actual data points are sufficient to reasonably fit a domain's D-CPT Law.

---

### Author Response · Authors · 2024-08-10
**Summarization on the Responses**

Thanks for handling/reviewing our submitted manuscript: "D-CPT Law: Domain-specific Continual Pre-Training Scaling Law for Large Language Models". We would like to thank the reviewers for their insightful and constructive comments and suggestions. By addressing each of the issues raised by the reviewers, we believe that the quality and clarity of our D-CPT Law can be improved a lot. The major responses are summarized as follows:

(1). We have carefully discussed the advantages and reasons for the choice of the formula of D-CPT Law (See Reviewer rpUb. Q1 and Reviewer sFHX. Q2).

(2). We have additionally compared the additional parameterizations of D-CPT Law and Cross-Domain D-CPT Law and discussed the advantages ours in detail (See Reviewer rpUb.Q1&Q2, Reviewer sFHX. Q2).

(3). We have discussed the advantages and reasons behind the choice of Qwen-1.5 series and the model sizes in detail (See Reviewer sFHX.Weakness).

(4). We have provided more clarification on the experimental settings of our main experiments (See Reviewer ycBX.Q2, Reviewer rpUb. Q3, and Reviewer sFHX.Q1).

(5). We have conducted additional experiments to compare the compute budget of grid search with the D-CPT Law, further demonstrating the efficiency of the D-CPT Law (See Reviewer ycBX.Weakness).

(6). We have provided more clarification on the reasons behind the design of D-CPT Law in the context of domain-specific continual pre-training (See Reviewer rpUb.Weakness).

(7). We have provided a more detailed analysis and clarification on the independent variables in the D-CPT Law (See Reviewer ycBX.Q1).

Again, we would like to sincerely thank you very much for these constructive comments and evaluation for our manuscript.

---

### Author Response · Authors · 2024-08-10
**Looking forward to feedback on the Responses**

Dear Reviewers:

Hello! Since the discussion period has started, we would like to kindly ask you to take a look at our responses and reevaluate our work based on our clarifications. Please let us know whether our response addresses your concerns or whether there is any further detail we can provide to help address them. We appreciate your time and consideration!

---

### Decision · Program_Chairs · 2024-09-25

**Decision:**

Accept (poster)

**Comment:**

This paper proposes a scaling law called D-CPT Law for domain-specific continual pre-training of large language models, which predicts model performance based on model size, dataset size, and data mixture ratio between general and domain-specific corpora. It has broad applicability to: trade-off between general and domain-specific abilities; find the optimal mixture for limited domain-specific data; allocate compute resources.

The paper extends the D-CPT Law to cross-domain settings with Cross-Domain D-CPT Law to predict performance on new domains with minimal additional training. The paper conducts experiments on 6 domains showing the effectiveness and generalizability of the proposed scaling laws.


**Strengths**
1) Addresses an important problem of optimizing data mixture for domain-specific LLM CPT training; 2) Proposes a novel scaling law with theoretical justification and empirical validation; 3) Demonstrates practical applications and generalizability; 4) Thorough experiments and analysis across multiple domains and settings.

**Weaknesses**
1) Limited to relatively small model sizes (0.5B-4B parameters) focused only on Qwen model series; Leaves some open questions about theoretical optimality of the chosen parameterization.


The paper is technically solid with good contributions. The authors provided reasonable justifications for their experimental choices regarding small models in the rebuttal. The authors provided good reasoning for their choices, and demonstrated some results on 7B models. Overall, the strengths outweigh the weaknesses.

Based on the reviews and discussion, I recommend accepting this paper (poster). Given the importance of the problem and the broad applicability, I recommend a spotlight